# Fundamental investigations on the sodium-ion transport properties of mixed polyanion solid-state battery electrolytes

Zeyu Deng [1] ✉, Tara P. Mishra [1,2], Eunike Mahayoni[3,4,5], Qianli Ma [6], Aaron Jue Kang Tieu[1], Olivier Guillon[6,7], Jean-Noël Chotard [3,4,5], Vincent Seznec [3,4,5], Anthony K. Cheetham [1,8], Christian Masquelier[3,4,5], Gopalakrishnan Sai Gautam [9] & Pieremanuele Canepa [1,2,10] ✉

Lithium and sodium (Na) mixed polyanion solid electrolytes for all-solid-state batteries display some of the highest ionic conductivities reported to date. However, the effect of polyanion mixing on the ion-transport properties is still not fully understood. Here, we focus on $Na_{1+x}Zr_2Si_xP_{3-x}O_{12}$ ($0 \leq x \leq 3$) NASICON electrolyte to elucidate the role of polyanion mixing on the Na-ion transport properties. Although NASICON is a widely investigated system, transport properties derived from experiments or theory vary by orders of magnitude. We use more than 2000 distinct ab initio-based kinetic Monte Carlo simulations to map the compositional space of NASICON over various time ranges, spatial resolutions and temperatures. Via electrochemical impedance spectroscopy measurements on samples with different sodium content, we find that the highest ionic conductivity (i.e., about 0.165 S cm$^{-1}$ at 473 K) is experimentally achieved in $Na_{3.4}Zr_2Si_{2.4}P_{0.6}O_{12}$, in line with simulations (i.e., about 0.170 S cm$^{-1}$ at 473 K). The theoretical studies indicate that doped NASICON compounds (especially those with a silicon content $x \geq 2.4$) can improve the Na-ion mobility compared to undoped NASICON compositions.

The reliability of rechargeable batteries and other energy storage devices depends on the fast delivery of ions between electrodes[1–4]. In rechargeable batteries, the safety and performance of electrolytes are as important as other battery components, such as electrodes. All-solid-state batteries, where liquid electrolytes are replaced by solid fast-ion conductors, offer a promising pathway for safer commercial lithium- and sodium- based batteries[4–6]. Several inorganic solid electrolytes,

such as the LiSiCON-type $Li_{4+x}Si_{1-x}Z_xO_4$ ($Z = P^{5+}$, $Al^{3+}$, $Sn^{4+}$ and/or $Ge^{4+}$)[7,8], $Li_{10}MP_2S_{12}$ based on $Li_4MS_4$:$Li_3PS_4$ ($M = Ge^{4+}$, $Sn^{4+}$, and $Si^{4+}$)[9–16], and $Na_{1+x}Zr_2Si_xP_{3-x}O_{12}$ (hereafter referred as NASICON)[17–19] leverage mixed polyanion frameworks to attain some of the highest ionic conductivities (≈10 mS/cm at -293 K) reported so far[4].

Ion transport in mixed polyanion solid electrolytes has been investigated previously using both computational and laboratory

[1]Department of Materials Science and Engineering, National University of Singapore, Singapore 117575, Singapore. [2]Singapore-MIT Alliance for Research and Technology, 1 CREATE Way, 10-01 CREATE Tower, Singapore 138602, Singapore. [3]Laboratoire de Réactivité et de Chimie des Solides (LRCS), CNRS UMR 7314, Université de Picardie Jules Verne, 80039 Amiens, Cedex 1, France. [4]RS2E, Réseau Français sur le Stockage Electrochimique de l'Energie, FR CNRS 3459, F-80039 Amiens, Cedex 1, France. [5]ALISTORE-ERI European Research Institute, FR CNRS 3104, Amiens F-80039 Cedex 1, France. [6]Forschungszentrum Jülich GmbH, Institute of Energy and Climate Research, Materials Synthesis and Processing (IEK-1), 52425 Jülich, Germany. [7]Helmholtz-Institute Münster, c/o Forschungszentrum Jülich GmbH, 52425 Jülich, Germany. [8]Materials Department and Materials Research Laboratory, University of California, Santa Barbara 93106 California, USA. [9]Department of Materials Engineering, Indian Institute of Science, Bengaluru 560012 Karnataka, India. [10]Chemical and Biomolecular Engineering, National University of Singapore, 4 Engineering Drive 4, Singapore 117585, Singapore. ✉e-mail: msedz@nus.edu.sg; pcanepa@nus.edu.sg

experiments[4,7,9,11,18]. Yet transport properties measured with different techniques typically vary by orders of magnitude[20]. On the one hand, averaged transport properties obtained from experimental measurements, such as electrochemical impedance spectroscopy, solid-state nuclear magnetic resonance, and quasi-elastic neutron scattering[4,6,16,21], may incorporate effects arising from defects and grain boundaries[22]. On the other hand, classical or ab initio molecular dynamics (MD) studies perform "one shot" simulations on selected bulk structures, which may not be sufficiently representative in terms of longer time-scale transport processes as well as in the number of possible arrangements of different polyanions[2,3,23–25]. Thus, there is a need to reconcile the experimental measurements and simulations to guide the development and manufacturing of the next generation of inorganic solid-state electrolytes for secondary battery applications.

In this study, we have selected the NASICON $Na_{1+x}Zr_2Si_xP_{3-x}O_{12}$ ($0 \leq x \leq 3$) electrolyte as an example to elucidate the role of polyanion mixing on the macroscopic transport properties, including ionic diffusivity and conductivity. The choice of NASICON is justified by the large body of data available on this system since its discovery more than 40 years ago[17,18,26–29], making it easier to reconcile previous experimental and computational studies.

To capture the large statistical variance in transport properties introduced by mixed polyanions in NASICON samples, we developed a high-fidelity kinetic Monte Carlo (kMC) model that captures the accuracy of density functional theory (DFT) calculations. More than 2,000 distinct kMC simulations served to map the statistically vast compositional space of $Na_{1+x}Zr_2Si_xP_{3-x}O_{12}$ over a long-time range—in the realm of milliseconds—and spatial resolution, with varying temperature.

First, our model reproduces existing measurements of Na-ion transport in NASICON, suggesting that a robust sampling of both the spatial and temporal axes is required in simulations to accurately estimate these properties. The reproduction of measured Na-ion transport in NASICON also implies that our model correctly captures the collective nature of Na-ion transport, which is responsible for the high ionic conductivity observed in NASICON. Second, our simulations elucidate the impact of the thermodynamic forces driving the random distribution of $PO_4^{3-}$ and $SiO_4^{4-}$ groups in NASICON during synthesis (and subsequent thermal treatment) on the Na-ion transport. Third, our statistical insights can guide the selection of optimal doping and thermal treatment strategies to further improve the properties of NASICON electrolytes. For example, our analysis suggests that higher $Na^+$ conductivity in NASICON can be achieved by increasing the content of $SiO_4^{4-}$ units in place of $PO_4^{3-}$ moieties, while maintaining high Na content in the structure. Motivated by these computational findings, we have also synthesized and characterized selected compositions of the NASICON to validate the theoretical analysis. These findings, demonstrated on NASICON electrolytes, are general and transferable to the study of ionic transport in other topical mixed-polyanion Li and Na solid electrolytes.

## Results

Figure 1a depicts the crystal structure of NASICON, where red $SiO_4/PO_4$ tetrahedra share corners with two blue $ZrO_6$ octahedra forming the "lantern units". In the rhombohedral high-temperature NASICON structure (space group $R\bar{3}c$), there are two independent and partially occupied sodium sites, Na(1) and Na(2). The partial occupancy of these sites gives rise to the high ionic conductivity in NASICON. A third interstitial Na site, known as Na(3), adjacent to Na(1) and Na(2) has been previously considered[30,31]. However, Zhang et al.[32] showed that Na(3) is negligibly occupied over a wide range of temperatures ($200 < T < 800\,°C$) and less relevant for Na-ion transport in NASICON. The structural images in Fig. 1b, c emphasize the Na(1) sites, with their neighboring Na(2) (orange spheres) and Si/P (red spheres) sites, resulting in distorted hexagonal prisms (silver). For visualization purposes, Fig. 1b, c omit the Zr and O atoms.

Figure 1d shows the smallest relevant unit—the migration unit—encompassing all possible migration events for Na ions in NASICON. The migration unit centered around Na(1) highlights the connectivity between the Na(1) site and its closest six Na(2) neighbor sites, as well as the neighboring Si(P) atoms. Each migration unit (see Fig. 1b, c) is edge-shared with six neighboring migration units, where each unit can exhibit its own distribution of Na/vacancies and Si/P. In the NASICON structure, Na-ion transport must always involve both Na(1) and Na(2) sites[18,30,33,34], giving rise to migration pathways of the kind Na(1) ⟷ Na(2). The specific Na-ion migration pathway depends entirely on the Na and Si/P content within the structure[19,30,31], which sets the occupations and the relative stabilities of Na(1) and Na(2) sites. For example, at low Na content ($x = 0$), Na ions reside only in Na(1) sites, favoring a Na(1) ⟷ Na(2) ⟷ Na(1) pathway[17,30,32]. At higher Na content ($x \geq 2$), both Na(1) and Na(2) sites are occupied to varying degrees, and any of the two migration pathways can be active[19,30]. For macroscopic Na diffusion to occur in NASICON, several successful migration hopping events, between adjacent migration units, need to happen, for which it is critical that Na atoms can hop from Na(1) to Na(2) sites (or vice versa) within each unit. Thus, we considered a single migration event to be Na(1) ⟷ Na(2) within a single migration unit, in the presence of varying Na and Si/P contents.

For any Na composition in the range $0 \leq x \leq 3$, the concentrations of Si, and P are set by the NASICON stoichiometry, such that the composition guarantees global charge neutrality (i.e., the NASICON cell is electrostatically neutral). However, each migration unit can exhibit random occupancies of Si/P in Si/P sites, and Na/vacancies in the Na(2) sites (panels (b–d) of Fig. 1). Consequently, the migration of Na ions may occur through migration units that are not locally charge neutral, i.e., a given unit by itself may not be electrostatically neutral. In our work, we consider all possible Na(1)↔Na(2) hopping events within a given migration unit, including scenarios with and without local charge neutrality.

We evaluated all the Na-ion migration pathways that are possible within a migration unit of Fig. 1d using the nudged elastic band method (NEB)[35] together with density functional theory (DFT), and using the strongly constrained and appropriately normed (SCAN) meta-generalized gradient approximation (Supplementary Note 1)[36]. The NEB method is designed to calculate migration barriers in inorganic solid-state electrolytes, as highlighted by prior studies[2,3,37,38]. We identified distinct 32 Na-hops, in the migration unit of Fig. 1d, which included barriers computed at different Si and P compositions and Na/vacancy contents (Supplementary Note 2). The calculated migration barriers at 0 K are defined as the difference between the highest energy state and the configuration with the lowest energy—the initial and/or final end point[37]. The validity of our computed barriers is supported by the data available in the literature. For example, our calculated barrier at $NaZr_2(PO_4)_3$ (~458 meV, Supplementary Table 1) agrees well with existing experimental values (~470 meV)[26] and ab initio MD simulations (~472 meV)[32]. The computed NEB barriers used to generate the heatmap in Fig. 1e are shown in Supplementary Fig. 1, Supplementary Fig. 2, and Supplementary Fig. 3.

Typically, migration events have an intrinsic directional dependence if the initial and final states are not equivalent, i.e., a Na(1) → Na(2) hop may exhibit a different migration barrier compared to the reverse hop, Na(2)→Na(1), due to energy differences between the end points, namely, Na(1) and Na(2). Van der Ven et al.[39] introduced the kinetically resolved activation (KRA) barrier, or $E_{KRA}$ (see Eq. (2) in the "Methods" section), which removes the directional dependence of the migration barriers. Note that low values of $E_{KRA}$ correspond to low values of migration barriers and vice versa. Furthermore, we trained a local cluster expansion (LCE) Hamiltonian[40] on the computed $E_{KRA}$ values (listed in Supplementary Table 1), which assigns a single $E_{KRA}$ value per migration unit and enables simulating large supercells of NASICON over long-time scales within our kinetic Monte Carlo (kMC)

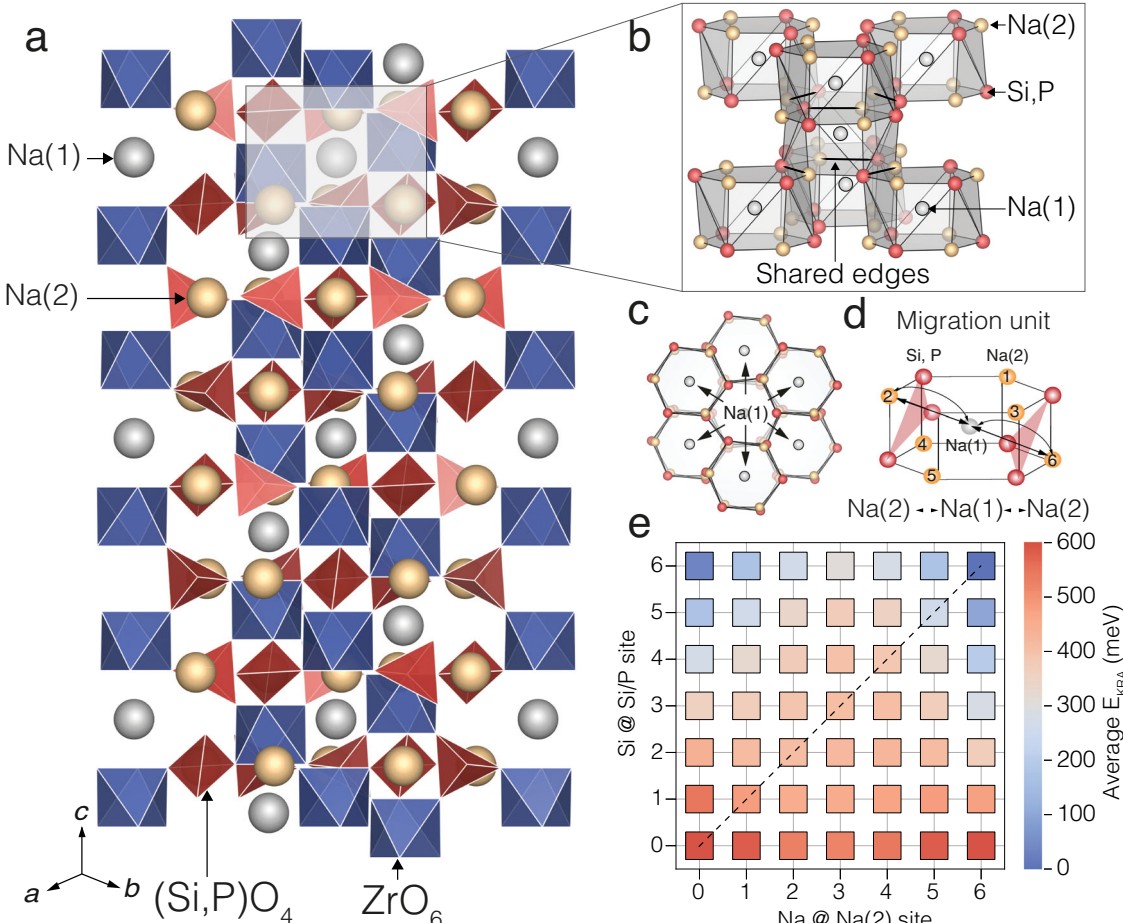

**Fig. 1 | Characteristics of sodium-ion transport in NASICON.** Structural models of $Na_{1+x}Zr_2Si_xP_{3-x}O_{12}$ (**a–d**) and Na-ion migration barriers (**e**). In (**a–d**), the Na(1) sites are indicated by silver spheres, the Na(2) by orange spheres, the (Si/P)O$_4$ groups by red tetrahedra, Si/P atoms by red spheres, and ZrO$_6$ units by blue octahedra. b and c depict the local environment of Na(1), with each Na(1) surrounded by six neighboring Na(2) atoms (orange spheres) and six Si/P (red spheres) atoms. For simplicity, O and Zr atoms are not shown in (**b–d**). Each silver hexagonal prism in (**b**) or (**c**) represents the first coordination shell of a Na(1) site. Panel (**d**) is the migration unit used to study Na-ion migration in NASICON, and Na must hop across several

different migration units to ensure Na diffusion. Red triangles in (**d**) indicate the bottlenecks caused by Si/PO$_4$ tetrahedra (oxygen atoms are not shown). (**e**) shows the averaged kinetically resolved activation (KRA) barriers for Na(2)⟷Na(1) hops, with varying Na(2) site occupation and Si/P content per migration unit. The barriers were extracted from a local cluster expansion model, which was fitted to the calculated nudged elastic band (NEB) barriers (Supplementary Table 1). The diagonal line in (**e**) shows locally charge neutral Si/P configurations. The computed NEB barriers used to generate the heatmap in (**e**) are available in Supplementary Fig. 1, Supplementary Fig. 2, and Supplementary Fig. 3.

package ("Methods" section and Supplementary Note 3 and Supplementary Note 4). The quality of the trained LCE to predict the computed NEB migration barriers is demonstrated in Supplementary Fig. 4, whereas the LCE formalism is detailed in Supplementary Eq. (1), Supplementary Eq. (2), and Supplementary Eq. (3). The main advantage of the LCE over computationally expensive NEB calculations at all possible configurations is that it enables the rapid estimation of $E_{KRA}$ (and the corresponding migration barriers) at all Na(2), Si and P compositions in a given migration unit (whose coordinates are given in Supplementary Table 2). Supplementary Table 3, Supplementary Table 4, and Supplementary Table 5 report the values of point terms, pairs, triplets and empty clusters, forming the LCE model Hamiltonian.

The average $E_{KRA}$ values obtained by the LCE model are plotted in Fig. 1e, with the dashed black diagonal indicating migration units that are locally charge neutral. Off-diagonal barriers are for locally charged situations and reflect multiple Na and Si/P configurations. For example, there are 400 possible configurations for a migration unit with 3 Na and 3 Si, and the $E_{KRA}$ of ~410 meV (Fig. 1e) is obtained by averaging the barriers across all the configurations. A similar averaging procedure is used for all scenarios where multiple Na/vacancy and Si/P configurations can exist within the same migration unit.

Depending on the Si/P environments, two distinct regions of high and low average $E_{KRA}$ are inferred from Fig. 1e. At lower Si or higher P content (bottom rows of Fig. 1e), high average $E_{KRA}$ values (~500 meV or similar) are observed. In contrast, increasing the Si content or decreasing the P content (top rows of Fig. 1e) lowers the average $E_{KRA}$ (values as low as ~50 meV). Importantly, an increase in Si concentration in NASICON lowers the average $E_{KRA}$. The monotonic decrease in $E_{KRA}$ (and migration barriers) with Si content can be attributed to lower $Si^{4+}-Na^+$ electrostatic repulsion compared to $P^{5+}-Na^+$ repulsion during Na-ion migration.

Changing the Na concentration at a given Si/P content within the migration unit does not result in a monotonic increase or decrease in average $E_{KRA}$. For instance, at 4/6 Si @ Si/P site in Fig. 1e, the average $E_{KRA}$ is low at both 0/6 Na @ Na(2) site and at 6/6 Na @ Na(2) site, with the average $E_{KRA}$ reaching a maximum at 3/6 Na @ Na(2) site. This non-monotonic behavior is likely due to a combination of local electrostatic repulsions and local charge imbalance within the migration unit. For example, at 6/6 Na @ Na(2) site and 0/6 Si @ Si/P site (bottom right corner of Fig. 1e), a high $E_{KRA}$ can be attributed to the higher electrostatic repulsion between the $Na^+$ and $P^{5+}$ ions. In contrast, low barriers at both 0/6 Na @ Na(2) and 6/6 Na @ Na(2) along 4/6 Si @ Si/P are

likely because of high local charge imbalance that destabilizes the initial/final Na(1) and Na(2) configurations along the pathway. Eventually, high Na content in the migration unit contributes to low $E_{KRA}$, particularly at high Si content, as indicated by the blue squares towards the top right corner of Fig. 1e. Indeed, the lowest values of $E_{KRA}$ are observed at both high Na and high Si contents.

Combining the LCE and kMC, we investigated the relevant Na-ion transport in NASICON, such as ionic diffusivity and conductivity. Supplementary Eq. (4), Supplementary Eq. (5), and Supplementary Eq. (6) of Supplementary Note 4 explain the working principles of kMC. Our first-principles calculations do not constrain the presence/absence of local distortion effects, and as a result our calculated migration barriers fully include such effects. The LCE, which is fitted to the calculated barriers is designed to coarse grain over all local distortions. Therefore, the kMC simulations (that rely on the LCE for calculating probabilities of hops) implicitly account for any local distortions within the structure.

The main advantage of using a kMC framework is that it gives access to statistically significant amounts of data over large length and time scales (demonstrated in Supplementary Fig. 5), as well as offering the ability to sample a wide variety of configurations. By comparison, a classical or ab initio MD simulation[3,23–25] performs a "one shot" representative calculation of a system at a given atomic configuration and composition, over smaller length and orders of magnitude shorter time scales.

In total, more than 2000 kMC simulations were performed to predict the Na⁺ diffusivities and conductivities at selected temperatures of 373, 473 and 573 K of 50 NASICON models (with varying Si/P configurations) at 11 distinct Na concentrations. A preliminary canonical Monte Carlo simulation at 1500 K that explores the thermodynamic landscape of $Na_{1+x}Zr_2Si_xP_{3-x}O_{12}$ at the desired composition, before each kMC simulation, is required to create model structures with a random distribution of silicate and phosphate units, as observed in existing structural reports (X-ray/neutron diffraction experiments) of NASICON. Previously, Deng et al. have demonstrated that Si/P redistribution upon cooling is hindered due to the high migration energy barriers (~4.02 eV) of $SiO_4$ and $PO_4$ units[19]. Using canonical Monte Carlo simulations from previous research works[19], 550 starting NASICON configurations (50 configurations × 11 concentrations) were generated at a high temperature (~1500 K), which mimics the typical synthesis conditions and thermal treatments of $Na_{1+x}Zr_2Si_xP_{3-x}O_{12}$ and other NASICON-based electrodes[17,28,30,33]. Using high-temperature structures as starting configurations in our kMC guarantees a random distribution of $PO_4$ and $SiO_4$ tetrahedra in the NASICON structures, as reported in the literature[17,30,32,33].

The choice of temperatures for the study of Na-ion transport in the present research work (measured at 373–573 K) was largely motivated by the availability of experimental measurements for comparison, but our framework can be easily extended to other temperatures. The comprehensive statistical analysis of this study addresses the role played by the $PO_4$ and $SiO_4$ distributions on overall Na-ion transport in $Na_{1+x}Zr_2Si_xP_{3-x}O_{12}$[19].

Figure 2 shows the computed Na⁺ diffusivity (*D*, panel a), the conductivity (σ, panel b), Haven's ratio ($H_R$, panel c) and averaged

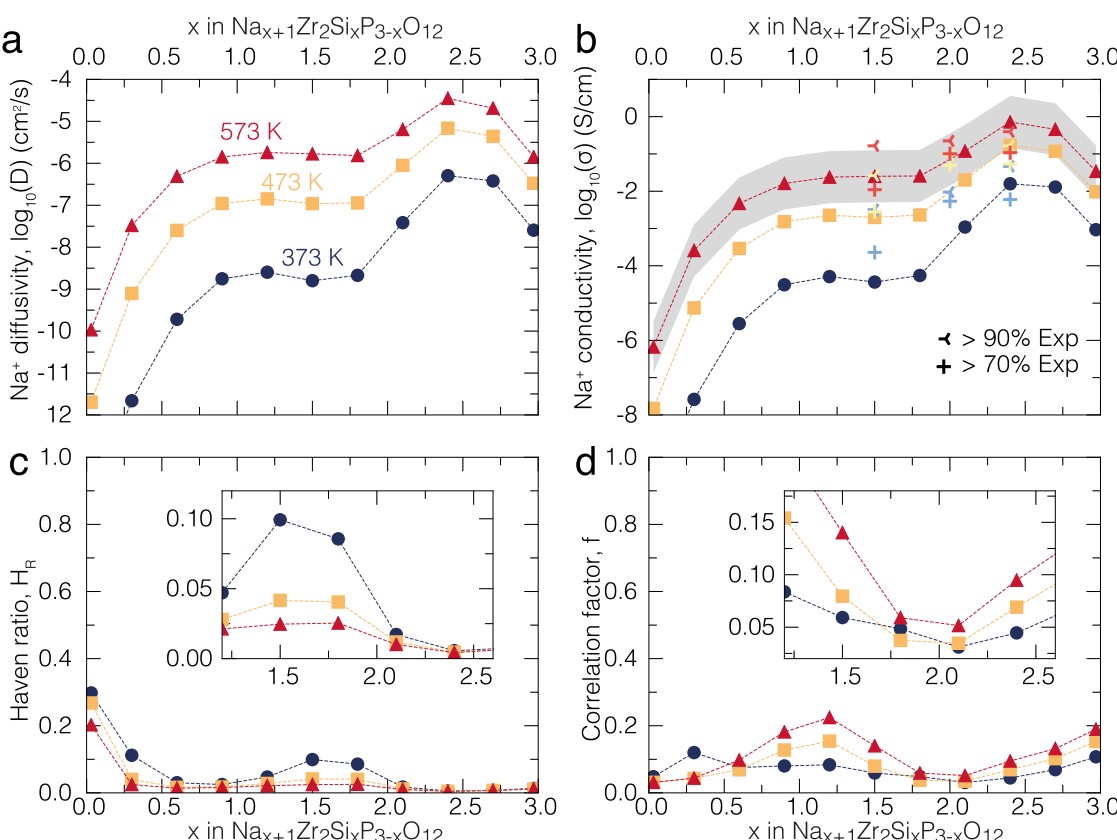

**Fig. 2 | Computed Na-ion transport properties of $Na_{1+x}Zr_2Si_xP_{3-x}O_{12}$ bulk based on kinetic Monte Carlo simulations.** Calculated Na⁺ diffusivity (**a**), conductivity (**b**), Haven's ratio (**c**) and averaged correlation factor (**d**) of $Na_{1+x}Zr_2Si_xP_{3-x}O_{12}$ at several temperatures: 373 (dark blue circles), 473 (orange squares) and 573 (red triangles) K, respectively. In panel (**b**), the computed ionic conductivities are compared with the experimental values of this work (Supplementary Fig. 6) at selected temperatures. Experimental values in (**b**) from this work are depicted with light blue (373 K), yellow (473 K), and red (573 K) crosses belonging to the same $Na_{1+x}Zr_2Si_xP_{3-x}O_{12}$ compositions but of pellets with different compacities (>70 and >90%, see legend). An exhaustive comparison with our and previous experimental values of ionic conductivities is shown in Supplementary Fig. 6. The gray band in panel (**b**) shows the interval of confidence of our predictions at 573 K, which is lower in magnitude than the variance of Na-ion conductivities observed experimentally (Supplementary Fig. 6).

correlation factor ($f$, panel d) at relevant temperatures of 373, 473, and 573 K, from kMC simulations of $Na_{1+x}Zr_2Si_xP_{3-x}O_{12}$ as the Na content ($x$) is varied. Note that the conductivity values ($\sigma$) in Fig. 2b were calculated using the Nernst–Einstein equation, as expressed in terms of diffusivity $D$ in Eq. (1)[3,23,24,39,40].

$$\sigma = \frac{e^2 C}{k_B T} D \qquad (1)$$

where $e$ is the elemental charge, $C$ is the concentration of Na ions, $k_B$ is the Boltzmann constant and $T$ is the temperature. Importantly, in Eq. (1), $D$ is the diffusivity of the center-of-mass of Na ions in the NASICON structure, which includes cross-correlations of the migrations of different Na ions (Supplementary Eq. (7) of Supplementary Note 4) and not the tracer diffusivity, as often used in many studies[23,39,40].

As confirmed by Fig. 2a, b, a temperature increase can significantly boost both Na⁺ diffusivities and conductivities across the full Na concentration. Interestingly, Na-ion diffusivity and conductivity increase for increasing Na (and Si) concentrations up to $x \sim 2.4$, beyond which there is a drop in both diffusivity and conductivity. Thus, we conclude that Na-ion transport in $Na_{1+x}Zr_2Si_xP_{3-x}O_{12}$ is fastest at $x \sim 2.4$, at all temperatures considered in this work, which agrees with our experimental observations presented below. Figure 2b also shows the experimental ionic conductivities of selected NASICON compositions from the same synthesis batch, but from pellets sintered at different compacities (>70% and >90%). Clearly, high compacity pellets provide higher Na-ion conductivities, especially in $Na_{2.5}Zr_2Si_{1.5}P_{1.5}O_{12}$. The conductivities obtained from the same samples but with different compacities (>70% and >90%) also serve as a guide to the variation to the experimental measurements.

Supplementary Fig. 6 reveals a spread of experimentally measured ionic conductivities from different reports[18,27,33,41–44], which is indicative of the difficulties in performing accurate measurements as well as the large variability in sample preparation, sampling of the compacity and level of impurities. Nonetheless, for $Na_3Zr_2Si_2P_1O_{12}$ our measured experimental conductivity of denser pellets (with 90% compacity) of ~0.165 S cm⁻¹ at 473 K is in excellent agreement with our computed value of ~0.170 S cm⁻¹. Our model also agrees with previous measurements[18,19,30,32] which report a low conductivity and diffusivity in $Na_1Zr_2P_3O_{12}$.

Previous studies have shown that correlated ion migration can contribute to high diffusivity and conductivity in ionic conductors[3,4,7,23,32,40,45,46]. In regimes of non-dilute diffusion carriers, the Haven's ratio ($H_R$) of Fig. 2c (Supplementary Eq. (8)) quantifies the degree of cross-correlation between migrating Na⁺ ions, i.e., the extent to which individual Na⁺ hops contribute to the overall motion of the Na⁺ center-of-mass. $H_R$ varies between 1 (no cross-correlation or a random movement of individual Na⁺ ions) and 0 (fully correlated transport or non-random movement). $H_R$ connects the diffusivity of Eq. (1) with the tracer diffusivity defined in Supplementary Eq. (9). While we observe small values of $H_R$ (0.01–0.2) throughout the Na concentration range in the NASICON, indicating significant cross-correlation between Na ions, $H_R$ at low Na concentrations ($x \sim 0$) is moderately high (~0.2) compared to high Na concentrations ($H_R \sim 0.01$ at $x \sim 3$). The variation in $H_R$ with Na composition ($x$) can be partly attributed to the occupancy of Na across the Na(1) and Na(2) sites in the NASICON. For example, at $x \sim 0$, Na⁺ mostly resides in the Na(1) sites, which are separated from each other by large distances (~6.47 Å), suggesting lower degrees of cross-correlations between migrating Na⁺ ions. At $x \sim 3$, Na⁺ ions populate both Na(1) and Na(2)[30], which are separated by shorter distances (~3.48 Å) than Na(1)–Na(1), possibly facilitating correlated migration of Na⁺ ions. Interestingly, cross-correlation between Na⁺ ions shows an increase with increasing temperature, as indicated by the drop in $H_R$ with

rising temperature at all $x$ (Fig. 2c). This trend also coincides with increasing of $D$ and $\sigma$, highlighting the positive impact of correlated motion on the overall Na-ion transport.

Another type of correlation, specifically between successive jumps of the same Na⁺ ion, is to estimate the average correlation factor, $f$, of Fig. 2d (see Supplementary Eq. (10)). $f$ measures the variation in local environments (resulting in a change of migration barriers) as experienced by individual Na⁺ ions as they migrate through NASICON. Furthermore, $f$ captures the deviation away from a truly random walk of a given Na⁺ ion within the structure, with $f = 1$ indicating a true random walk and $f = 0$ signifying a non-random walk. Thus, $f$ is different from $H_R$, which describes the correlation between the motions of different Na⁺ ions. Similar to our observations with $H_R$, we find that correlation effects, as measured by $f$, are quite pronounced at all Na concentrations in NASICON (Fig. 2d). For example, $f$ decreases progressively from 0.2 to 0.01 as $x$ varies from 1 to 2.

One of the main reasons behind the high degree of correlation (as indicated by $H_R$ and $f$) in NASICON is the high sensitivity of the $E_{KRA}$s to the local Si/P environment (Fig. 1e). Indeed, one may observe large variations in the local migration units, which lead to significant changes in $E_{KRA}$, at a given Na concentration. This is clearly shown in Supplementary Fig. 2, where the migration barrier varies between ~300 meV and ~800 meV at $x \sim 2$. Every Na⁺ migration event leads to a dynamic change in local environment by creating new vacancies and eliminating existing ones, which can have sizeable effects on the migration barriers (and hence $E_{KRA}$), eventually resulting in highly correlated motion.

It is worth analyzing how the distribution of Na ions across the Na(1) and Na(2) sites controls the Na-ion diffusivity (and conductivity) of NASICON. Figure 3a shows the averaged Na occupancy of the Na(1) and Na(2) sites, as extracted from our kMC simulations. We compare these values with single-crystal X-ray diffraction experiments by Boilot et al.[30] at specific compositions, and our previous grand-canonical Monte Carlo simulations[19], which do not account for the dynamics of the Na⁺ ions.

In NASICONs[18,47], Na-ion transport can only occur if multiple exchanges between Na(2) and Na(1) sites occur[18,30,31]. This implies that both Na(1) and Na(2) must be occupied to some degree[18,30,31] for facile ion transport. The Na site occupancies derived from our kMC data, which includes possible Na-ion migration events, at 443 K (170 °C, blue shapes in Fig. 3a) show that both Na(1) and Na(2) are partially occupied at compositions $1 < x < 3$. The kMC results are in qualitative agreement with X-ray measurements at 443 K of Ref. 30 (orange shapes in Fig. 3a). In contrast, occupancies generated by our previous grand-canonical Monte Carlo model (red filled shapes in Fig. 3a) indicate the equilibrium Na values, which are distinctly different from the kMC simulations (and experiments) that include the impact of Na movement.

Importantly, both our kMC and grand-canonical Monte Carlo simulations indicate that at $Na_1Zr_2P_3O_{12}$, Na⁺ ions will be almost exclusively located at Na(1), even after including Na⁺ dynamics at 443 K (Fig. 3a)[32]. In $Na_1Zr_2P_3O_{12}$, the Na(2) sites are empty because they are not thermodynamically stable at this composition[32,33]. Thus, the lack of Na atoms present in Na(2) sites that can act as diffusion carriers contributes significantly to the low $D$ (and $\sigma$) observed at low Na content ($x \sim 0$, Fig. 2), in addition to the high $E_{KRA}$ observed (i.e., at low values of Na@ Na(2) and Si@ Si/P in Fig. 1e).

By contrast, at high Na contents ($x \sim 3$, $Na_4Zr_2Si_3O_{12}$), Na⁺ diffusion is mostly limited by a lack of Na vacancies (either on Na(1) or Na(2) sites), which contributes to the moderately low computed $D$ and $\sigma$. Note that the computed $E_{KRA}$ barriers achieve their lowest values corresponding to high Na and Si content (Fig. 1e), which is similar to the local migration units that will be observed at $x \sim 3$, and should, in principle, contribute to increases in $D$ and $\sigma$. However, the sharp decrease in the number of available vacancies prevails over the decrease in $E_{KRA}$, eventually lowering D and σ. While we only consider

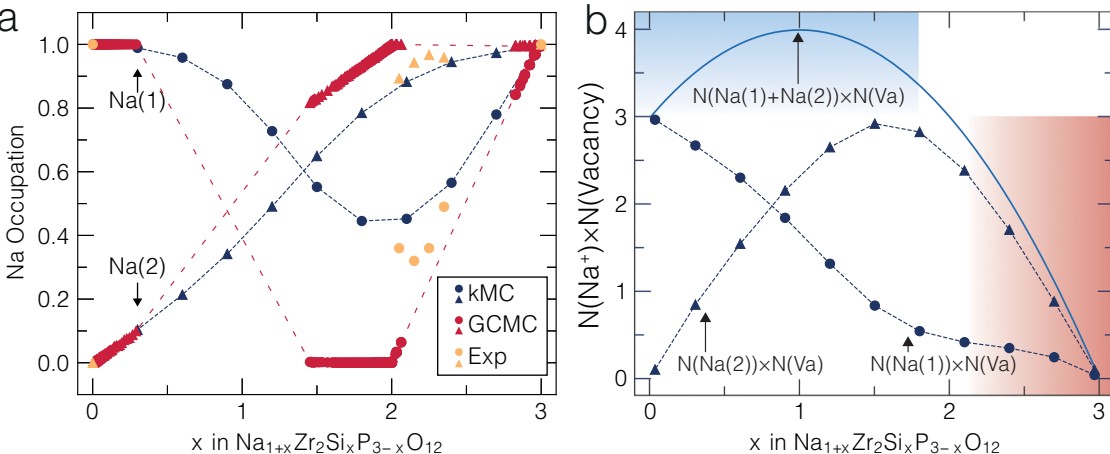

**Fig. 3 | Sodium site occupancies as a function of Na content.** Panel (**a**) shows the average Na site occupancy at the Na(1) site (blue circles) and the Na(2) site (blue triangles) from the kMC simulations at 443 K; they are compared with previous Grand Canonical Monte Carlo simulations[19] (red circles and triangles) at the same temperature, and existing experimental data[30] (in orange circle and triangles). Dashed lines are used as guide for the eye. Panel b shows N(Na$^+$) × N(Vacancy)

(solid blue curve, per f.u.) at 443 K from kMC simulations. The contributions to N(Na$^+$) × N(Vacancy) from individual sodium sites, Na(1) and Na(2) are shown as blue dashed lines. In panel (**b**), the shaded regions mark areas where Na-ion conductivity is highly (red) or slightly (blue) impacted by the charge carrier concentration.

vacancy-based migration mechanisms in the present research work, a recent study has also explored interstitial-based migration mechanisms in various solid electrolytes[48]. Given the robust agreement between our predictions and measurements, we believe that an interstitial-based mechanism is not active within NASICON structures.

Figure 3b further quantifies the extent of diffusion carrier availability within the NASICON structure. For example, the solid blue curve of Fig. 3b shows the product of the number of Na ions and vacancies in the structure [N(Na$^+$) × N(Vacancy)], which is a proxy for the availability of diffusion carriers in the structure, as a function of Na content. Figure 3b also splits the concentration of Na ions, N(Na$^+$), into site-specific quantities, N(Na(1)) and N(Na(2)), as indicated by the dashed blue curves.

In the blue-shaded region of Fig. 3b in the range $0 < x < 2$, N(Na$^+$) × N(Vacancy) is extended, signifying high availability of diffusion carriers, but the barriers for Na-ion motion are large as well (bottom half of Fig. 1e). In striking contrast, for $x > 2$, N(Na$^+$) × N(Vacancy) decreases sharply, along with a decrease in $E_{KRA}$ as well (top half of Fig. 1e), signifying an increase in $D$ and σ up to $x \sim 2.4$ followed by an eventual decline of both quantities as $x \rightarrow 3$. In Fig. 3a, at $x = 3$, both Na(1) and Na(2) sites are filled and no free charge carriers are available.

Experimentally and theoretically[17–19,30], at $x \sim 2.4$ both Na(1) sites and Na(2) sites are expected to be occupied to some degree and the Na$^+$ has enough vacancies to migrate. Furthermore, we expect the overall migration barriers to be low at $x \sim 2.4$ compared to $x < 2$, as quantified by trends in $E_{KRA}$ in Fig. 1e. Therefore, Na$_{3.4}$Zr$_2$Si$_{2.4}$P$_{0.6}$O$_{12}$ strikes the optimal combination of density of diffusion carriers (i.e., availability of Na ions and vacancies in both Na(1) and Na(2) sites), and low migration barriers to achieve the fastest Na-ion transport in the NASICON electrolyte.

To assess the reliability of our kMC calculations, we have synthesized different compositions of the NASICON, namely, Na$_{2.5}$Zr$_2$Si$_{1.5}$P$_{1.5}$O$_{12}$, Na$_3$Zr$_2$Si$_2$P$_1$O$_{12}$, and Na$_{3.4}$Zr$_2$Si$_{2.4}$P$_{0.6}$O$_{12}$ (Supplementary Table 6) and measured their Na-ion transport characteristics. Figure 4a shows the powder X-ray diffraction patterns (PXRD) measured at 298 K, with the PXRD profiles matching the rhombohedral (R$\bar{3}$c) Na$_{2.5}$Zr$_2$Si$_{1.5}$P$_{1.5}$O$_{12}$, the monoclinic (C2/c) Na$_3$Zr$_2$Si$_2$P$_1$O$_{12}$, and the monoclinic (C2/c) Na$_{3.4}$Zr$_2$Si$_{2.4}$P$_{0.6}$O$_{12}$ structures, respectively. Information about the synthesis of these NASICON compositions is given in the "Methods" section.

Rietveld refinement was used to extract the lattice parameters, atomic coordinates and Na occupancies of Na$_{2.5}$Zr$_2$Si$_{1.5}$P$_{1.5}$O$_{12}$, Na$_3$Zr$_2$Si$_2$P$_1$O$_{12}$ and Na$_{3.4}$Zr$_2$Si$_{2.4}$P$_{0.6}$O$_{12}$ (Table 1 and Supplementary Table 7, Supplementary Table 8 and Supplementary Table 9) the data obtained are in agreement with the state-of-the-art literature[17,49]. ZrO$_2$ was also detected in the NASCION samples (see Fig. 4a and Supplementary Table 7) as already reported in previous research works[30,33,50].

Powders of Na$_{1+x}$Zr$_2$Si$_x$P$_{3-x}$O$_{12}$ with $x = 1.5$, 2 and 2.4 were then sintered into pellets (see details in the "Methods" section and Supplementary Table 6) whose microstructures were analyzed with scanning electron microscopy (SEM). The SEM analysis of Supplementary Fig. 7, Supplementary Fig. 8, and Supplementary Fig. 9 suggests no appreciable changes in grain-size distributions across pellets of different Na$_{1+x}$Zr$_2$Si$_x$P$_{3-x}$O$_{12}$ compositions. The average particle sizes detected were 0.028, 0.043, and 0.035 µm$^2$ for $x = 1.5$, 2 and 2.4, respectively.

The variable temperature Na ionic conductivities are shown in the Arrhenius plots of Fig. 4b. Electrochemical impedance spectra at variable temperatures and their fitting details are also provided in Supplementary Fig. 10, and Supplementary Note 5. The high-temperature total ionic conductivities (Supplementary Fig. 10) of the synthesized Na$_{1+x}$Zr$_2$Si$_x$P$_{3-x}$O$_{12}$ phases were fitted with appropriate equivalent circuits (see the "Methods" section). The extracted activation energies for all three NASICON compositions were obtained from the Arrhenius plots (ln(σT) vs. 1000/T) shown in Fig. 4b. For the composition $x = 1.5$, the extracted activation energy is ~0.39 eV, and no phase transition is observed due to the stability of the rhombohedral phase at room and higher temperatures. The compositions $x = 2.0$ and $x = 2.4$ show two linear domains with different activation energies displayed in Fig. 4b. Thus, the Arrhenius plots of Fig. 4b reproduce the documented monoclinic-to-rhombohedral phase transition at ~433 ± 15 K upon heating for the compositions Na$_3$Zr$_2$Si$_2$P$_1$O$_{12}$, and Na$_{3.4}$Zr$_2$Si$_{2.4}$P$_{0.6}$O$_{12}$[51–53]. The activation energies for $x = 2.0$ are ~0.34 eV (below the phase transition ~431 K) and ~0.28 eV (above ~431 K), which are in good agreement with previous reports[33,54–60]. Na$_{3.4}$Zr$_2$Si$_{2.4}$P$_{0.6}$O$_{12}$ shows similar activation energies, with values of ~0.32 eV (below ~414 K) and 0.24 eV (above ~414 K), in agreement with the literature data[54]. The composition with $x = 1.5$ has the lowest ionic conductivity for all temperatures compared to structures with $x = 2.0$ and 2.4. Note that low-temperature electrochemical impedance measurements of the monoclinic Na$_{3.4}$Zr$_2$Si$_{2.4}$P$_{0.6}$O$_{12}$ phase indicate high Na-ion

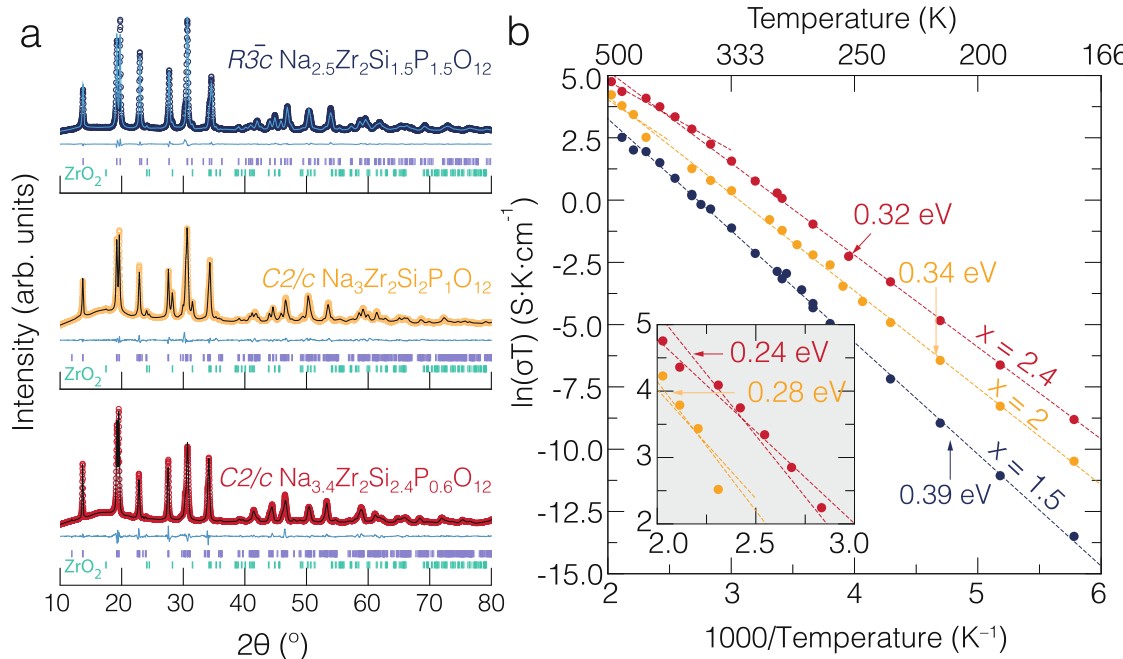

**Fig. 4 | Powder X-ray diffraction results for selected NASICON compounds at 298 K and their ionic conductivities measured using pellets.** In panel **a**, powder X-Ray diffraction patterns (dark blue, orange, and red circles) and Rietveld refinements (solid lines in cyan, dark brown and dark red) for $Na_{1+x}Zr_2Si_xP_{3-x}O_{12}$ (x = 1.5, 2.0, and 2.4) are shown. **b** Arrhenius plots from electrochemical impedance measurements of three samples (x = 1.5 in dark blue, x = 2.0 in orange, x = 2.4 in red). Complex impedance plots (i.e., Nyquist plots) are shown in Supplementary Fig. 10, while Supplementary Fig. 11 shows the total, grain boundary, and bulk conductivities of the NASICON samples.

conductivity, as also confirmed by high-frequency and low-temperature AC impedance data (see following paragraphs and Supplementary Fig. 10 and Supplementary Fig. 11)[58]. The ionic conductivity of x = 2.0 showed that the measured values of this work are within the range of reported values[18,27,33,41–44].

Because sintered pellets of NASICONs were used to measure Na-ion transport, their total ionic conductivities depend on resistance contributions arising from both the bulk and the grain boundaries. Therefore, we have performed low-temperature (between 173 and 433 K) and high-to-moderate frequency (in the range 3 GHz to 1 Hz) AC electrochemical impedance spectroscopy measurements (see details in the "Method" Section) of the three nominal compositions, x = 1.5, x = 2.0 and x = 2.4 to separate the effect of grain boundary contributions on the total ionic conductivity of the pellets. These measurements are shown in Supplementary Fig. 10. At low temperatures (e.g., ~193 K for x = 3.4, Supplementary Fig. 10c) NASICON samples show two predominant semicircles in the complex impedance plot (also called

Nyquist plot). For example, in Supplementary Fig. 10c there is a smaller semicircle occurring at low resistances and high frequencies, which was modeled by an resistive (R) element in parallel with a constant phase element (CPE) representing bulk Na-ion transport. The CPE element describes the non-ideal capacitor behavior of solid-state conductors. This is followed by a larger semicircle at higher resistances and lower frequencies which was modeled by another R+CPE element representing Na grain-boundary resistance. The bulk and grain boundary features of the impedance responses are indistinguishable at higher temperatures (see Supplementary Fig. 10). Two main considerations stem from the analysis of these electrochemical impedance spectra:

1. In NASICON samples with x = 1.5 and 2.4, the bulk ionic conductivities are always greater than or similar to those of the grain boundaries. For composition x = 2 (see Supplementary Fig. 10b), the separation of the grain boundary and bulk conductivities is different, with the grain boundary conductivity being slightly higher than the bulk conductivity at the lowest temperatures.

2. A close inspection of the Arrhenius plots of all NASICON compositions (Supplementary Fig. 11a and Supplementary Fig. 11c) demonstrates that the grain boundary conductivity (resistance) always approaches the value of the bulk conductivity at temperatures ≥ 333 K and for composition x = 1.5 and x = 2.4. In the case of $Na_3Zr_2Si_2P_1O_{12}$ the grain-boundary conductivity is always higher than the bulk (Supplementary Fig. 11b). This observation indicates that at moderate-to-high temperatures the effect of grain boundaries on Na-ion transport is negligible and the total conductivity is solely controlled by Na-ion transport in the NASICON grains.

For the NASICON compositions x = 2.0 and 2.4, Fig. 4b shows a monoclinic-to-rhombohedral phase transition above ~413 K, which is accompanied by a sudden change of activation energy (slope). However, from Fig. 4b $Na_{2.5}Zr_2Si_{1.5}P_{1.5}O_{12}$ remains rhombohedral across the whole range of temperature investigated. Because of this linearity in

## Table 1 | Lattice parameters of selected NASICON samples obtained from PXRD at about 298 K

| Space group | $Na_{2.5}Zr_2Si_{1.5}P_{1.5}O_{12}$ $R\bar{3}c$ | $Na_3Zr_2Si_2P_1O_{12}$ $C2/c$ | $Na_{3.4}Zr_2Si_{2.4}P_{0.6}O_{12}$ $C2/c$ |
|---|---|---|---|
| a | 8.9918 (6) | 15.7344 (4) | 15.6844 (15) |
| b | 8.9918 (6) | 9.1001 (2) | 9.0493 (9) |
| c | 22.9983 (19) | 9.1990 (2) | 9.2320 (7) |
| β | – | 124.3397 (13) | 124.157 (4) |
| Volume | 1610.36 | 1087.59 (5) | 1084.30 (17) |
| Z | 6 | 4 | 4 |
| Volume/Z | 268.39 | 271.9 | 271.08 |

Lattice constants, angles and volumes are in Å, °, and Å³, respectively. Z is the number of formula units. Supplementary Table 7, Supplementary Table 8, and Supplementary Table 9 contain the refined coordinates and relative occupancies of the three samples as obtained from Rietveld refinements.

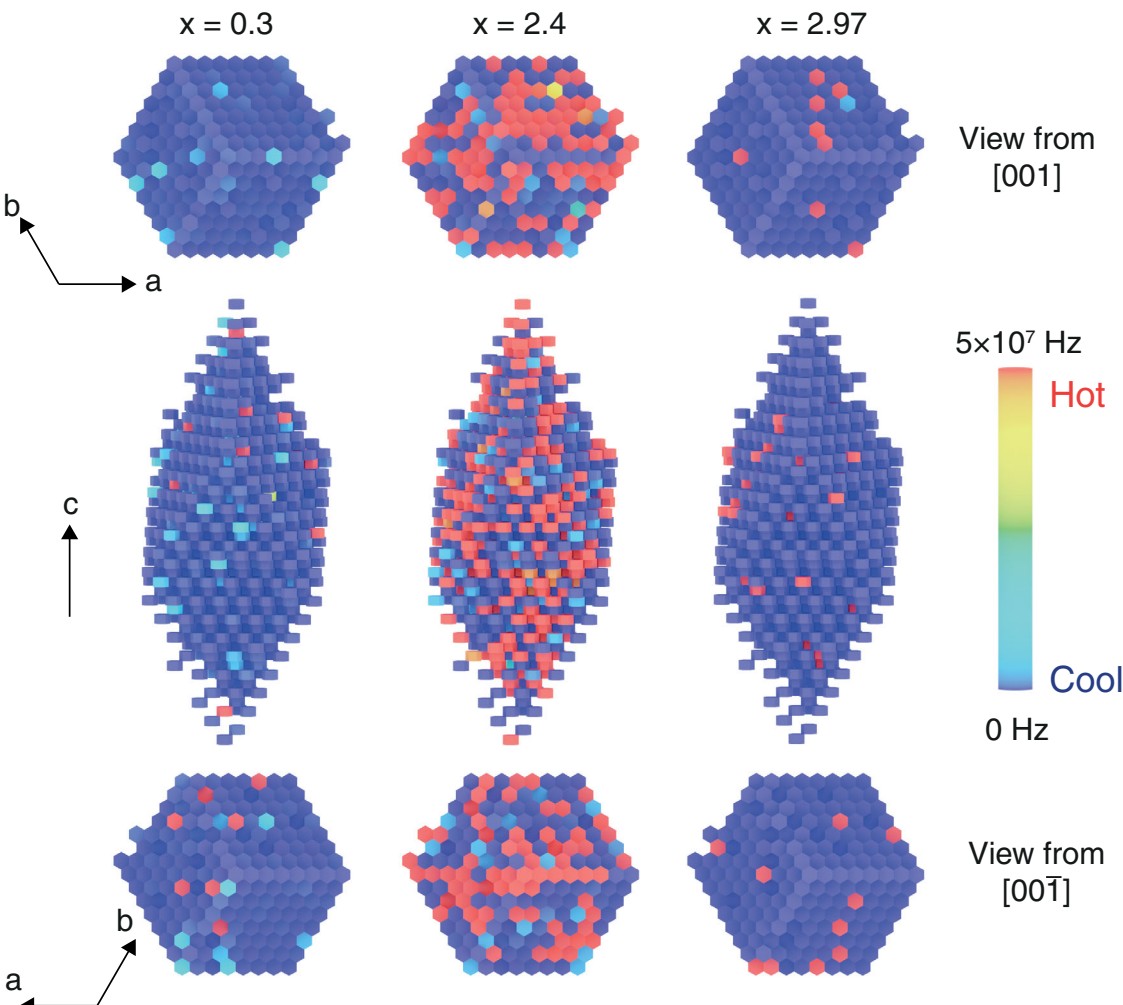

**Fig. 5 | Sodium hopping frequencies in NASICON materials.** Heatmap representation of the frequency (in Hz) of Na-ion migration from selected kMC simulation supercells (8×8×8 of the primitive cell, including 1024 migration units) at Na contents of $x$ = 0.3, 2.4 and 2.97. The temperature in all panels is 573 K. Each colored hexagonal prism represents a migration unit, as exemplified in Fig. 1d. The hopping frequency of each migration unit is taken by averaging the probability of the six possible migrations of Na⁺ between the central Na(1) site and the six Na(2) sites in each unit. The top and bottom panels show views along the $c$-axis from above and below the supercells.

---

$Na_{2.5}Zr_2Si_{1.5}P_{1.5}O_{12}$, we could safely extrapolate the grain boundary resistance and bulk conductivity at high temperature from the low-temperature and high-frequency measurements, as demonstrated by Supplementary Fig. 12. This analysis demonstrates that at high temperatures the effects of grain boundary resistance on the total conductivity are negligible, with Na-ion transport being entirely dominated by the ionic conductivity of the bulk.

## Discussion

Experimentally, ionic diffusivity can be measured by a number of techniques, such as solid-state nuclear magnetic resonance, quasi-elastic neutron scattering and secondary-ion mass spectroscopy[20,61], while AC electrochemical impedance spectroscopy is the method of choice for determining ionic conductivity in inorganic solid electrolytes[62]. Indeed, we have used AC electrochemical impedance spectroscopy in this study. Although classical and ab initio molecular dynamics simulations have proved invaluable in the prediction of ionic conductivities in solid electrolytes[2,3,7,20,22,24,40,45,63–66], we have demonstrated that first-principles-based kMC can access significantly longer time scales in the millisecond range and larger length scales (a total of 21,504 atom supercells in the present work), enabling us to approach

experimental time and space resolutions, and establish a robust link between measurements and theory.

Our kMC simulations provide the hopping frequencies of Na⁺ migration, at different Na concentrations and temperatures, which can be efficiently evaluated and mapped in large supercell models. The hopping frequency is defined in Supplementary Eq. (4) and includes the attempt frequency and the migration barrier. From our kMC data, we plot heatmaps indicating the spatial migration of Na ions in Fig. 5, along with the frequency associated with each Na migration event. Figure 5 shows the hopping frequency of three representative NASICON compositions, i.e., $x$ = 0.3, 2.4, and 2.97, capturing Na-ion transport in low, intermediate, and high sodium concentrations. Each panel in Fig. 5 contains 1024 migration units (Fig. 1e), where each migration unit is represented by a distinct colored hexagonal prism. In Fig. 5, the migration units with high hopping frequencies, shown by red prisms, imply that Na ions have a high probability of migrating beyond the migration unit, thus contributing actively to macroscopic diffusion (the inference is vice versa for blue/purple prisms corresponding to low hopping frequencies). As expected, and as shown in Fig. 2, $x$ = 2.4 shows a significantly higher hopping frequency compared to compositions $x$ = 0.3 and 2.97.

The direct visualization of the hopping frequency of $Na^+$ migration is insightful for unraveling the respective effects played by $PO_4$ and $SiO_4$ moieties in the overall $Na^+$ transport. In Fig. 5, at low Na concentrations, e.g., $x = 0.3$, most migration units appear purple (low frequency) signifying low diffusivity/conductivity, whereas these blue regions turn progressively to yellow and eventually red for increasing values of $x$. At $x \sim 2$, most Na ions are actively engaged in ion transport as most migration units are red (Fig. 5), highlighting facile ionic transport, which is consistent with the highest values of observed $D$ and $\sigma$ at $x \sim 2.4$ (Fig. 2). As $x$ approaches 3 ($x \sim 2.97$ shown in Fig. 5), the color of the migration units eventually become purple, coinciding with the drop in $D/\sigma$ at high Na contents (Fig. 2) due to the low availability of Na vacancies. This informative analysis can be extended to other topical solid electrolytes with mixed polyanion sulfides, such as $GeS_4$, $PS_4$ and $SiS_4$[8,9,12–14]. The analysis of Fig. 5 can also be relevant for the rationalization of complex total scattering experiments obtained, for example by neutron-based techniques.

The underlying structural models to generate the data in Fig. 5 are based on disordered NASICON structures, where at a given Na concertation the $SiO_4$ and $PO_4$ units are randomly distributed. This is also the case for the most researched sulfide electrolytes for all-solid-state lithium-based batteries with various degree of mixing of $GeS_4$, $SiS_4$, $SiS_4$ and $SnS_4$ moieties[9,11–14]. Thus, these NASICON structures truly reflect the experimental synthesis conditions (~1200 °C for solid-state synthesis[18,33]) and/or heat treatments (e.g., low-temperature sol–gel synthesis[27,67] followed by sintering at ~1200 °C for densification[18,33,67]), and mimic the disorder of the phosphate and silicate units accessed at high temperatures. Previously we have shown that, under equilibrium conditions, NASICON should phase separate into P-rich and Si-rich domains, particularly across Na concentrations from $x = 0–2$[19]. It is also important to understand the impact of phase separation (or lack thereof) on Na-ion diffusivity and conductivity.

We have extended our kMC model to structures produced in regimes of complete phase separation (see Supplementary Fig. 13 and Supplementary Fig. 14). Notably, the computed ionic conductivities and other related quantities predicted in regimes of phase separation look similar to those reported in Fig. 2, with the exception of a dip in $D$ and $\sigma$ around $x \sim 1.8 – 2$ compared to the scenario of fully disordered system (non-phase separated). However, this dip in ion transport properties approximately accounts for one order of magnitude, well within the error of the available experiments and our calculations, thus making it challenging to detect the signature of phase separation via diffusivity/conductivity measurements or calculations. Our calculations suggest that Na-ion diffusivity and conductivity cannot be used to identify the underlying phases within the NASICON electrolyte. We believe this conclusion may also be valid for other solid electrolytes and electrodes of interest, particularly those adopting the NASICON structure, which can also thermodynamically favor phase separation[68,69]. Notably, heatmaps produced in structures with phase separation (Supplementary Fig. 14) do clearly depict the presence of phase boundaries (differentiating Si-rich and P-rich domains), indicating that robust structural characterization is still the best way of detecting phase separation.

Our computed barriers and kMC simulations suggest that higher $Na^+$ conductivity in NASICON can be achieved by increasing the $SiO_4$ to $PO_4$ ratio. There are two reasons for this: (i) higher $Si^{4+}$ content decreases that of $P^{5+}$ cations in the system, thereby reducing the electrostatic repulsion between the $Na^+$ and $P^{5+}$ cations during Na migration, and (ii) due to the longer Si–O bonds in $SiO_4$ compared to the P–O bonds in $PO_4$, Si-rich migration units will be larger, facilitating migration of large ions, such as $Na^+$. Thus, tuning the Si concentration (or, equivalently, increasing the number of 4+ instead of 5+ cations within the polyanion groups) in the NASICON structure (via aliovalent doping, for example), provides a new way for optimizing ionic

transport in this class of ionic conductors. For NASICONs with higher Si content, Na off-stoichiometry achieved by "stuffing" excess sodium in the material might provide increased Na mobility. This strategy is commonly applied in many ion conductors[44,70]. Indeed, the last ~40 years of research on NASICON materials has investigated aliovalent doping as a way to improve the Na-ion conductivity of this material. An increase of Na ions is typically achieved by subvalent doping of the $Zr^{4+}$ with trivalent $Al^{3+}$, $Cr^{3+}$, $Dy^{3+}$, $Er^{3+}$, $Fe^{3+}$, $Gd^{3+}$, $In^{3+}$, $Sc^{3+}$, $Tb^{3+}$, $Y^{3+}$, and $Yb^{3+}$, and/or divalent $Co^{2+}$, $Mg^{2+}$, $Ni^{2+}$, and $Zn^{2+}$ species[26,28,32,71–73]. Combining a high Si content in the structure with trivalent doping on the $Zr^{4+}$ site can potentially improve Na-ion transport at high Na concentrations in NASICONs ($x\rightarrow3$). Alternatively, supervalent doping on the $Zr^{4+}$ site with $V^{5+}$, $Bi^{5+}$, $Sb^{5+}$, $Nb^{5+}$, and $Ta^{5+}$ cations is used to increase the number of Na vacancies, and also to reduce (by ~30 K) the temperature of the monoclinic-to-rhombohedral phase transition[74]. Supervalent doping is also a viable strategy for increasing the Si content in the NASICON structure while preserving the stoichiometric ratio of Na ions and vacancies. Specifically, the ionic radii of $Sb^{5+}$ (~0.60 Å), $Nb^{5+}$ (~0.64 Å) and $Ta^{5+}$ (~0.64 Å) in octahedral environments appear compatible with that of $Zr^{4+}$ (~0.72 Å)[75]. The higher stability of the 5+ oxidation state of the three candidates represents the most viable choice for incorporating more Si into the NASICON structure. Furthermore, it has been shown that isovalent doping of $Ge^{4+}$ on $Si^{4+}$ can also increase the size of the migration unit[76], potentially facilitating Na-ion diffusion.

In summary, we have used a combination of state-of-the-art computational tools to explain the influence of polyanion mixing, composition and temperature on ionic transport in $Na_{1+x}Zr_2Si_xP_{3-x}O_{12}$. This was achieved by using a combination of density functional theory-based nudged elastic band calculations, a local cluster expansion Hamiltonian, and kinetic Monte Carlo simulations over millisecond time scales and nanometer length scales. Our study has demonstrated that low migration barriers for Na ions can be achieved by having local environments that are rich in Si and Na, which can be attributed to the lower electrostatic repulsion between $Si^{4+}$ and $Na^+$ (vs. $P^{5+}$ and $Na^+$) during Na-ion migration. A further increase of Si content in $Na_{1+x}Zr_2Si_xP_{3-x}O_{12}$ can be achieved by doping Zr with stable 5+ cations, such as Ta, Nb, Sb, while maintaining an optimal Na composition. Importantly, we discovered a complex interplay between migration barriers and diffusion carrier availability in determining the overall ionic diffusivity/conductivity. For example, at low Na concentrations ($0 < x < 2$), high migration barriers dominate, resulting in a gradual increase in diffusivity/conductivity with increasing $x$. At high Na concentrations ($x > 2.5$), the lack of diffusion carriers dominates (despite low migration barriers), resulting in a drop in diffusivity/conductivity. Thus, we observe that a Na composition of $x \sim 2.4$ is optimal in terms of combining enough diffusion carriers with low migration barriers, resulting in the highest diffusivity/conductivity. Through careful synthesis and AC impedance characterization of the $Na_{3.4}Zr_2Si_{2.4}P_{0.6}O_{12}$, we have confirmed its high Na-ion conductivity, as predicted by our simulations. Using $Na_{1+x}Zr_2Si_xP_{3-x}O_{12}$ as a model system, we demonstrated the importance of sampling statistically vast composition, length, and time scales to capture the highly correlated ionic motion that can be observed in mixed polyanion systems. Our findings are significant for the optimization of mixed polyanion solid electrolytes, such as sulfide-based systems which can achieve regimes of super-ionic conductivity—typical of solid-state systems with ionic conductivities $\geq 10^{-3}$ S cm$^{-1}$ at 298 K.

## Methods
### Sodium migration barriers from density functional theory
Na migration barriers were calculated using the nudged elastic band (NEB) method[35] through density functional theory (DFT) simulations as implemented in the Vienna ab initio simulation package (VASP)[77,78]. All the calculation parameters used in the DFT simulations can be found in

Supplementary Note 1 of the SI. The NEB barriers were modeled at three representative Na concentrations, at $x = 0$, 2, and 3 as in $Na_{1+x}Zr_2Si_xP_{3-x}O_{12}$. This selection is motivated by previous knowledge of the compositional phase diagram of $Na_{1+i}Zr_2Si_xP_{3-x}O_{12}$[19], where NASICON exhibits three distinct ground states at 0 K. Note, the rhombohedral-to-monoclinic phase transition that is typical of NASI-CON is the result of disorder (rhombohedral)/order (monoclinic) transformations on the Na/Vacancy and Si/P sublattices. Since the DFT calculations describe accurately phase changes at different Na compositions[19], our structural and computational model fully captures these order/disorder situations depending on conditions (temperature, composition).

Special care is required for the NEB barrier calculations of $NaZr_2(PO_4)_3$ ($x = 0$) which follows a pathway Na(1)→Na(2)→Na(1). In $NaZr_2(PO_4)_3$ all the Na(1) sites are occupied and Na-ion migration is only possible if vacancies in Na(1) are introduced. For this specific case the energy required for $Na^+$ migration should also include the formation energy of Na vacancy, which we computed using the method of Ref. 79. We found that the Na vacancy formation energy in $NaZr_2(PO_4)_3$ is ~474 meV, which we added to our model. Whereas the Na vacancy formation energy at i = 2 and 3 is negligible (-13 meV). Note that at $x = 2$, Na(1) and Na(2) are only partially occupied which guarantees facile Na transport.

To remove the directional dependence of migration barriers, we define the kinetically resolved activation (KRA) barrier, $E_{KRA}$ of Eq. (2):

$$E_{KRA} = E_{barrier}[Na(1) \longleftrightarrow Na(2)] - \frac{1}{2}(\Delta E_{end}) \qquad (2)$$

where the $E_{barrier}[Na(1) \longleftrightarrow Na(2)]$ is the NEB barrier[39,40], and $\Delta E_{end}$ is the absolute difference between the computed energies of the initial and final end point structures. The distribution of $E_{KRA}$ at different configurations of Na and Si/P environments is shown in Supplementary Fig. 1, Supplementary Fig. 2, Supplementary Fig. 3, and Supplementary Table 1.

## Local cluster expansion hamiltonian
The calculated $E_{KRA}$ are fitted to a local cluster expansion (LCE) Hamiltonian built around a migration unit (Fig. 1d and reference coordinates in Supplementary Table 2) centered on the Na(1) site and using a cut-off radius of 5 Å. Details of the LCE model and the fitting strategy are given in the Supplementary Information. The fitted Hamiltonian can be used efficiently to compute migration barriers inside the migration unit at any given Na/vacancy and Si/P content/configuration. The LCE Hamiltonian includes 1 point, 5 pair, and 1 triplet terms (Supplementary Table 3, Supplementary Table 4 and Supplementary Table 5), respectively. The LCE Hamiltonian can reproduce the migration barriers with an RMS error of ~±38 meV. The robustness of the LCE Hamiltonian was cross-validated using the leave-one out method. Notably, a variability of ~±38 meV corresponds to a less than an order of magnitude in diffusivity (± 60 meV)[37], which is the typical uncertainty of experimental measurements and theoretical calculations for electrode materials[40].

## Kinetic Monte Carlo simulations
We implemented a rejection-free kinetic Monte Carlo (kMC) simulation scheme in an in-house-code, as described in Refs. 39, 40. The initial configurations for the kMC were generated using the canonical Monte Carlo simulation (based on our previous work where we constructed a global cluster expansion model on the NASICON)[19] in an 8 × 8 × 8 supercell at $0 \le x \le 3$ containing 4096 distinct Na sites and 3072 Si/P sites. These structures were generated at 1500 K, and close to the experimental synthesis temperature[17,28,34]. For each of the 11 compositions sampled ($x = 0.03, 0.3, 0.6, 0.9, 1.2, 1.5, 1.8, 2.1, 2.4, 2.7, 2.97$),

50 structural models were generated using the canonical Monte Carlo approach[19].

We used kMC to evaluate the transport properties of NASICON crystals, including $Na^+$ diffusivity, bulk conductivity, Haven's ratio, and correlation factor. These physical quantities are averaged for each of the 50 initial configurations per Na composition. During the kMC simulation, only Na ions can hop while the remaining atoms are frozen. Each kMC simulation included 2,048,000 equilibration steps, followed by 12,288,000 sampling steps for statistical analysis at 373, 473 and 573 K, respectively. In total, we have run 1650 independent kMC simulations (11 compositions × 50 initial configurations × 3 temperatures) with a total of ~14 million steps per configuration, which totals to approximately 23 billion kMC steps. For investigating the impact of phase separation on ionic transport, we followed the same procedure as above, except the canonical Monte Carlo, equilibrated structures were generated at 573 K, instead of at 1500 K.

## Synthesis and structural characterization of $Na_{1+x}Zr_2Si_xP_{3-x}O_{12}$ compounds
Synthesis of $Na_{1+x}Zr_2Si_xP_{3-x}O_{12}$ for $x = 1.5$ and $x = 2.0$ were carried out using a sol–gel method with $NaNO_3$ (Alfa Aesar, 99%), $NH_4H_2PO_4$ (Sigma Aldrich, 98%), $ZrO(NO_3)_2.xH_2O$ (Alfa Aesar, 99%), $Si(OC_2H_5)_4$ (Tetraethyl orthosilicate/TEOS, Rectapur, 99%) and citric acid (Alfa Aesar, >99%) as precursors followed by carefully optimized thermal treatments. Citric acid was first mixed with 300 mL of ethanol (VWR, > 99.8%):deionized water (1:1), at 343 K for 1 h. Then a stoichiometric amount of the precursors was introduced to the solution and the temperature was raised to 363 K for 1.5 h. The resulting gel was then dried for 8 h, at 453 K. A first calcination was carried out in open air at 873 K (starting from 296 K) for 8 h in Nabertherm P330 muffle furnace, followed by natural cooling in air. Using the same furnace, the sample underwent a second calcination in open air at 1373 K (starting from 296 K) for 18 h, followed by natural cooling in air. For the composition $x = 2.4$, a solvent-assisted solid-state reaction method was applied[54]. Corresponding amounts of $NaNO_3$ (VWR, 99.7%), and $ZrO(NO_3)_2$ (Sigma Aldrich, 99%), were dissolved into deionized water. A stoichiometric amount of $Si(OC_2H_5)_4$ (Merk, 99%) was also added to the solution while stirring. When $Si(OC_2H_5)_4$ was hydrolyzed, the corresponding amount of $NH_4H_2PO_4$ (Merk, 99%) was added to the system while stirring. The homogeneous aqueous system then changed to a mixture of a colloidal sol and precipitates of complex zirconium oxyphosphate compounds. The whole mixture was dried at ~358 K. The dried powder was calcined at ~1073 K for 3 h in air. After calcination, a white powder was obtained. The calcined powder was then milled in ethanol (Schmittmann GmbH with concentration of 97%). with zirconia balls on a custom-made milling bench for 48 h and dried at ~343 K for 12 h. All samples were handled in air.

X-Ray diffraction patterns of the obtained white powders were then measured at ~298 K using a Bruker D8 diffractometer with copper source ($K_{\alpha 1} = 1.54056$ Å and $K_{\alpha 2} = 1.54439$ Å), and a step size of 0.021°. Rietveld refinements were done using WinPLOTR and FullProf software. For $x = 1.5$ the pattern was fitted using the R$\bar{3}$c space group[18,76,80,81], while $x = 2.0$ and $x = 2.4$ were fitted using the C2/c space group[33,82].

## Electrochemical impedance measurements of the $Na_{1+x}Zr_2Si_xP_{3-x}O_{12}$ samples
All samples were handled in air. For the preparation of impedance measurements, the three NASICON samples were cold pressed into pellets of 10 mm diameter via a hydraulic press, with an applied pressure of 4 ton and 10 min hold prior to sintering. Subsequently, for the $x = 1.5$ and 2.0 samples, using a Nabertherm P330 muffle furnace the pellets were sintered in open air at ~1473 K for 18 h. For the $x = 2.4$ composition, the pellet was sintered in open air at 1558 K for 6 h (in a Nabertherm P330 muffle furnace). The sintered pellets were then gold-

sputtered (~75 nm) with a sputter coater (Ted Pella Inc., Cressington 108 auto). Vacuum / Ar charge was repeated 3 times before sputtering. 20 mA was applied as the beam-current. The sputtering lasted 120 s for each side of the sintered pellets. A representative gold coated NASICON sample is shown in Supplementary Fig. 15.

Carbon paper (papyex) was added on each side of the pellet to ensure good contacts for the impedance measurements. For the high-temperature measurements, we employed a cell immersed in a Janis STVP-200-Sol Cryostat connected to a Solartron 1260 A/Solartron 1296 dielectric interface. The impedance cell, the heating elements and a thermocouple are part of the Janis STVP-200-Sol setup, whose relevant parts are displayed in Supplementary Fig. 16. The electrodes are made of gold, supported by a stainless steel framework (Supplementary Fig. 16a–d). The impedance cell is constituted by a two electrode setup, and the cell is symmetric: Au|NASICON|Au. The working and counter electrode are made of gold. Impedance spectra were sample in the frequency range from 10 MHz to 1 Hz (20 points per decade of frequency), with an alternating voltage amplitude of 20 mV. Variable temperature measurements were recorded, where the sample was cooled down every 20 K from 600 K to 253 K (with 15 min holding time), using a Lakeshore 335 temperature controller.

For high-frequency impedance measurements, two commercial electrochemical systems (Keysight E4991B and Novocontrol Technologies Alpha-A) with an AC frequency range from 3 GHz to 1 MHz and from 10 MHz to 1 Hz (20 points per decade of frequency swept) were applied, respectively. An alternating voltage amplitude of 20 mV was used during measurements. The impedance cell setup used for the high-frequency measurements is displayed in Supplementary Fig. 17. The impedance cell is constituted by a two electrode setup, and a symmetric cell Au|NASICON|Au was used. Both working and counter electrode are made of gold. The temperature dependent impedance was recorded between 433 K and 173 K in a temperature-controlled chamber (Novocontrol Technologies BDS1100). The data analysis and fitting of the impedance data for both high and moderate frequency were performed with the Zview software (Scribner Associates Inc.).

A description of the appropriate equivalent circuits to fit the AC impedance is provided in Supplementary Note 5. The equivalent circuits to fit the impedance spectra at high temperatures are reported as inset in Supplementary Fig. 10d–f, and for low temperatures as inset of Supplementary Fig. 10a–c. The parameters obtained (and their relative errors) from fitting selected impedance spectra (at 197 K of Supplementary Fig. 10a–c) are reported in Supplementary Table 10.

### Scanning electron microscopy measurements and microstructure analysis

The microstructure of the same sintered pellets used for the electrochemical impedance measurements (prior gold sputtering) was analyzed using the secondary electron mode in a scanning electron microscope (SEM, FEI Quanta 200F) which enables us to identify the particle-size distribution. Sintered samples were handled and transported in open air and placed on a SEM sample holder (in open air). The incident electrons of the SEM measurements were carried out using a 10 kV accelerating voltage. The detection of grains from the micrographs of the sintered pellets was achieved thorough an adaptive mean thresholding algorithm, as implemented in the openCV python library[83]. The detection of the local thresholding values was achieved using a neighborhood area of $15 \times 15$ px$^2$. To detect individual $Na_{1+x}Zr_2Si_xP_{3-x}O_{12}$ particles, two morphological operations were carried out in sequence, namely, erosion followed by dilation using a minimal kernel of $3 \times 3$ px$^2$. The preceding operation results in the values of pixels associated with grain boundaries to be 255 in a GRY scale. Using this information, the $Na_{1+x}Zr_2Si_xP_{3-x}O_{12}$ particles could be isolated from their grain boundaries. Using the scikit-image python

package a statistical analysis of the grain size was conducted[84], which is reported in Supplementary Fig. 7, Supplementary Fig. 8 and Supplementary Fig. 9.

### Reporting summary

Further information on research design is available in the Nature Research Reporting Summary linked to this article.

## Data availability

All the computational data associated with this study, including the NASICON model structures used in the kMC calculations are available on Zenodo: https://doi.org/10.5281/zenodo.6827359.

## Code availability

The kMC code utilized for this study is available at https://github.com/caneparesearch/NASICON_KMC_paper_data.

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

## Acknowledgements

P.C., C.M., A.K.C., E.M., V.S., and J.-N.C. are grateful to the ANR-NRF for the funding of the NRF2019-NRF-ANR073 Na-MASTER project. P.C. and Z.D. acknowledge funding from the National Research Foundation under NRF Fellowship NRFF12-2020-0012. T.P.M. was supported by the National Research Foundation (NRF) Singapore through Singapore MIT Alliance for Research and Technology (SMART)'s Low Energy Electronic Systems (LEES) IRG. The computational work was performed on resources of the National Supercomputing Centre, Singapore (https://www.nscc.sg). A.K.C. thanks the Ras Al Khaimah Center for Advanced Materials for financial support. We thank Prof. Stefan Adams at the National University of Singapore and Dr. Theodosios Famprikis at TUDelft for fruitful discussion.

## Author contributions

P.C. designed and supervised the project. Z.D., T.P.M., and P.C. performed the NEB simulations and fitted the local cluster expansion Hamiltonian with discussions with G.S.G. Z.D. developed all the tools for constructing models, Hamiltonian fitting, the kinetic Monte Carlo code and the data analysis. E.M. and Q.M. carried out the synthesis of the NASICON phases, E.M., Q.M., O.G., A.J.K.T., and T.P.M. performed the PXRD, and the impedance measurements in Germany and Singapore. E.M. and A.J.K.T. performed the data analysis of the experimental data under the supervision of Q.M., O.G., J.-N.C., V.S. and C.M. Z.D., T.P.M., and P.C. wrote the first draft. Z.D., P.C., G.S.G., and A.K.C. contributed to the data analysis of the computed data. All the authors contributed to the discussion and final version of this manuscript.

## Competing interests

The authors declare no competing interests.
