## [Peer Review File · Nature Communications]

REVIEWER COMMENTS

Reviewer #1 (Remarks to the Author):

In this manuscript, Deng et al. report a theoretical and experimental study on the ionic transport in mixed polyanion solid electrolytes. Specifically, the study focuses on the effect of silicon and phosphorous content on the ionic transport of the NASICON material $\text{Na}_{1+x}\text{Zr}_2\text{Si}_x\text{P}_3-x\text{O}_{12}$.

From a theoretical perspective, the authors utilized the approach of Ben Morgan (reference 47 of the manuscript) to obtain insights on the dynamic ion correlations that take place in the various compositions of the NASICON electrolyte studied. Although the sheer number of calculations performed is a great feat of its own merit, I am not sure how innovative it is. That being said, since I am not a theoretician myself, I cannot fully gauge the extent of the complexity and challenges in producing such a dataset, and perhaps missed something in which the authors pushed the boundaries further than what was reported by Ben Morgan.

The measurement and analysis of the presented impedance results is not correct. It is unreasonable that for impedance measurements using ion-blocking electrodes the impedance spectra do not start at the origin (0,0) of the Nyquist impedance plot. The main example for this error is on Figure 4C, for the spectrum at 373 K, in which the blue semicircle extends from 25 Ohms to ca. 145 Ohms. Since the authors mention that they measured up to frequencies of 30 MHz, there is no reason for all the spectra to not start at the origin. A possible source for the offset is that the electrolyte is reacting with the gold contact materials. The formation of such Na-Au alloys is possible at the temperatures reported here, so the authors should check if the electrodes have been affected by the measurement conditions. The other explanation for the offset of the blue semicircle is that this corresponds to the grain boundary resistance of the sample. Since the samples tested are not dense (the authors report pellet densities in the range of 70-77%), it is likely that grain boundary resistances are the main contributor to the impedance spectra. If the explanation for the offset is the second one, then it is a big problem for the manuscript because the authors make direct comparisons of the theoretically obtained ionic conductivities and those obtained experimentally. Such a direct comparison is only possible with fully dense samples, impedance data single crystals or when the impedance data is analyzed considering grain (bulk) and grain boundary resistivities.

Since the impedance data analysis is flawed and it serves to support one of the key findings of the manuscript, namely “we have demonstrated that first principles- based kMC can access significantly longer time scales in the millisecond range and larger length scales (a total of 21,504 atom supercells in the present work), enabling us to approach experimental time and space resolutions and establish a robust link between measurements and theory”, I cannot recommend the manuscript for publication as is.

Reviewer #2 (Remarks to the Author):

In this manuscript, the authors report the theoretical systematic study of the Na NASICON family by a novel kinetic Reverse Montercalo approach to assessing the impact of PO₄ substitution by SiO₄ polyanion. The authors also report experimental work on different compositions to validate their computation predictions. Overall, the work is well executed and of interest to the wider battery materials community and readership of Nature Communications, however, this reviewer suggests the below points to be addressed before publication of this work in Nature Communications:

1) The authors should discuss the limitation of their method for long-range diffusion analyses since the method does not incorporate the effect of grain boundaries. This is of particular interest when trying to understand experimental results where the material's microstructure can dominate the transport properties. Specifically in their work, where the authors compare their bulk (intra-grain) predicted transport properties with their experimental results, which seems to include both, bulk and grain boundaries (inter-grain) conductivity results. The authors could measure their transport properties at lower temperatures, which usually helps to decouple bulk and grain boundary resistances to Na⁺ transport. This would be highly important, given that the authors have employed different synthetic methodologies and obtained different relative densities, which impacts the total conductivity reported. SEM images should be included to compare if microstructures are still comparable.

2) The reported kMC method only allows for Na⁺ to move, constraining the rest of the atoms to fix positions. Considering that for other polyanion systems have been reported polyanion rotations during Na⁺ diffusion – so-called paddle-wheel effect, do the authors anticipate any limitation or their current methodology to this particular system or other sulphide systems where they expected to extend this methodology?

3) For the NBE method the authors have employed a temperature of 0 K, while with other methods and techniques temperatures above 300 K have been used. Do the authors expect a constant Arrhenius behaviour of the transport properties over the whole range of temperatures reported to make these results comparable?

4) This reviewer has some concerns regarding the temperature choices for their theoretical calculations. The authors use the structure obtained at 1500 K to mimic that of the maximum temperature employed

during experimental synthesis. In most cases, however, materials are cooled down to ambient temperature naturally, allowing the crystal structure to relax. This also seems to be the case of their experimental condition – no quenching protocol is mentioned. The authors should discuss this point and recalculate it as necessary. Additionally, the authors must perform Rietveld refinements, rather than Le Bail fittings, to obtain full structural information, importantly Na distribution within Na(1) and Na(2) sites, stoichiometry as well as quantification of ZrO₂ impurity presence. Particularly, the presence of this impurity is concerning in their Na = 3 sample, where a non-negligible amount of ZrO₂ is detected, which could affect the total transport properties of the material. Additionally, the authors' claim on phase purity of these compounds should be revised.

5) It would be helpful for the reader to include in the supporting information the structural parameters for the models employed. Considering these are quenched structures at 1500 K, which is the symmetry of their model used for the diffusion calculations? This is an important parameter considering the monoclinic to rhombohedral transition of the NASICON system at some compositions at temperatures below the quenching point selected. Additionally, Na⁺ occupancies across different sites could vary with temperature.

6) This reviewer also recommends the authors to calculate/show the windows sizes difference for Na⁺ diffusion when replacing P⁵⁺ with Si⁴⁺, and correlate these with their results/conclusions.

7) The authors should include in the abstract the temperature at which the reported conductivity is obtained. Furthermore, the authors claim high conductivity (page 28) of their sample with 0.1 mS cm⁻¹ at 200 °C, considering the current state-of-the-art in the field this claim should be revised.

8) The authors should also consider including the effect of the higher or optimal Na stoichiometry when discussing the positive effect of P⁵⁺ substitution with Si⁴⁺ on page 26.

As an additional comment of this reviewer, and out of the scope of the current work, this reviewer finds quite interesting and encourage the authors to investigate coupling their methodology to model the disorder or phase segregation with total scattering techniques to shed light on any clustering of this mixed polyanion systems that could escape conventional structural characterisation.

Dr Marco Amores, Austrian Institute of Technology

Reviewer #3 (Remarks to the Author):

In this paper, Deng et al. performed both experimental and theoretical simulations to elucidate Na ion diffusion properties both in bare $\text{NaZr}(\text{PO}_4)_3$ and Si doped on the P site in $\text{NaZr}(\text{PO}_4)_3$. This is a systematic study. However, I have a question in its novelty. There are many studies (hypothetical and experimental) available showing that additional Li or Na can be produced in many cathode materials [ex: Recently, Arunkumar et al. synthesized over-lithiated $\text{Li}_{2+x}\text{Ru}_{1-x}\text{CoO}_3$ cathode by aliovalent Co doping on Ru site in Li_2RuO_3 and concluded that there is an enhancement in the electrochemical lithium reversibility and Li^+ extraction compared to those associated in the pristine Li_2RuO_3 [Arunkumar, P., Jeong, W. J., Won, S. & Im, W. B. Improved electrochemical reversibility of over-lithiated layered Li_2RuO_3 cathodes: Understanding aliovalent Co^{3+} substitution with excess lithium. *J. Power Sources* 324, 428–438 (2016)]. This enhances the capacity and diffusion. Here, authors choose a phosphate based material to show that there is an agreement with theory and experiment.

Editor should decide whether this manuscript is considered or not. If yes, the following issues should be considered

Comment 1: Nonetheless, for $\text{Na}_3\text{Zr}_2\text{Si}_2\text{P}_1\text{O}_{12}$ our measured experimental conductivity (see Figure 4b) of $\sim 0.1 \text{ S cm}^{-1}$ at 573 K is in excellent agreement with our computed value of $\sim 0.2 \text{ S cm}^{-1}$. Clearly calculated value is overestimated by 100%. Is that excellent agreement?

Comment 2: Structures with charge-neutral vacancies at initial and final images in the diffusion path were fully optimized until the interatomic forces were less than 0.01 eV/\AA and stresses were less than 0.29 GPa.

Stress tensor looks high, not zero. Normally it should be 0.05 GPa or less than this.

Answers to Reviewer #1

Comment 1, Referee #1. In this manuscript, Deng et al. report a theoretical and experimental study on the ionic transport in mixed polyanion solid electrolytes. Specifically, the study focuses on the effect of silicon and phosphorous content on the ionic transport of the NASICON material $\text{Na}_{1+x}\text{Zr}_2\text{Si}_x\text{P}_{3-x}\text{O}_{12}$. From a theoretical perspective, the authors utilized the approach of Ben Morgan (reference 47 of the manuscript) to obtain insights on the dynamic ion correlations that take place in the various compositions of the NASICON electrolyte studied. Although the sheer number of calculations performed is a great feat of its own merit, I am not sure how innovative it is. That being said, since I am not a theoretician myself, I cannot fully gauge the extent of the complexity and challenges in producing such a dataset, and perhaps missed something in which the authors pushed the boundaries further than what was reported by Ben Morgan.

We thank the referee for this important comment. Undeniably, our manuscript and Ref. 47 (DOI: 10.1098/rsos.170824, later addressed as Dr. Morgan) rely on kinetic Monte Carlo (kMC) simulations, but with many notable differences. The text below and **Table 1** establish that our methodology is significantly different in terms of accuracy, general applicability, and reliability than that proposed previously by Dr. Morgan. Please find a summary below:

1. Although both our methodology and Dr. Morgan's method rely on a lattice model, the two methods calculate the ionic transport in completely different ways. Dr. Morgan's methodology treats the motion of Na^+ (or Li^+) in a simplistic "ion-gas" fashion neglecting the crucial interactions between the diffusing cations (Na^+ or Li^+) and the anion/cation species of the framework. **In contrast, our method fully incorporates the interaction between Na-ions and the host framework.**
2. Dr. Morgan's model relies on empirical pair potentials (and assigns arbitrary parameters), which are not derived for the system investigated. Therefore, the results presented in Ref. 47 are not quantitative in nature, and hence not comparable directly to experiments. In contrast, our methodology derives a many-body "potential", trained on density functional theory (DFT) calculations, which can accurately capture 'long' range interactions and correlations between the migrating Na-ions and the host framework. **Our methodology can provide quantitative data comparable directly with experimental measurements** (see main text); **we consider this a significant advancement of the kMC methodology** compared to Dr. Morgan's method.
3. Dr. Morgan's methodology has been tailored around a garnet-type structure, only capturing the disorder on the Li/Vacancy sub-lattice, which lacks generality. We have considered disorder on both the Na/Vacancy and Si/P sub-lattices. **Thus, our methodology is versatile as it can be applied to systems displaying disorder on multiple sub-lattices.** The broader applicability of our methodology is certainly a major aspect of our strategy.

In summary, our method is superior and carries more novelty in many ways as that proposed by Dr. Morgan, and we hope Referee #1 will be convinced of this.

Table 1 Comparison between our method and Dr. Morgan's method (DOI: 10.1098/rsos.170824).

Observable	This work	Dr Morgan Ref. 47
Accuracy	Results are computed from first-principles DFT simulations and directly comparable to experiments.	Results are not quantitative and highly dependent on empirical parameters (e.g., E_{nn} and E_{site}), whose values are set arbitrarily. Results are not directly comparable to experiments.
Diffusion barrier & conductivity	Diffusion barrier are calculated in a many-body fashion (from a local cluster expansion model). The many-body model was in turn trained on accurate DFT data.	Empirical definition of activation barriers via a gas model.
Cation-anion Interactions	Interactions between Na^+ and vacancies, as well as the host framework (silicate/phosphate/ Zr^{4+}) are fully included in the local cluster expansion model.	Only the interaction between Li^+ (or Na^+) and vacancies are incorporated.
Hamiltonian	A many-body type expansion, including point, pair, triplet, etc. interactions, without any loss of generality is developed. This model is extracted from accurate DFT data and bears the same accuracy of first-principles calculations.	Empirical pair-potentials are used. This is an extremely crude treatment of the Li^+ (or Na^+) and vacancy interactions. Note, these methods were popular several decades ago when DFT data was difficult to obtain.
Effects of distortion and environment	The local environment experienced by the Na migration is entirely captured with the help of a local cluster expansion. The effect of tetrahedra and octahedra distortions are implicitly considered in the calculated activation barriers.	A rigid model is used that neglects the description of local distortion effects, caused by ion migration. This is an oversimplification of the method.
Generality of the method	The method is highly general and can be applied to a wide range of solid electrolytes and ionic conductors, with disorder on multiple sublattices Here we have shown the case of NaSICON with two distinct disordered sublattices, i.e., Na/Vacancy and Si/P.	Only suited to the garnet structure with specific octahedral and tetrahedral sites (Li/Vacancy only).
Treatment of sublattice disordering	Disordering of both Na/Va and Si/P is fully considered through a general statistical analysis. Indeed, this analysis can be extended to more species and sub-lattices.	Only the disordering of Li/Va is considered

Comment 2, Referee #1. The measurement and analysis of the presented impedance results is not correct. It is unreasonable that for impedance measurements using ion-blocking electrodes the impedance spectra do not start at the origin (0,0) of the Nyquist impedance plot. The main example for this error is on Figure 4C, for the spectrum at 373 K, in which the blue semicircle extends from 25 Ohms to ca. 145 Ohms. Since the authors mention that they measured up to frequencies of 30 MHz, there is no reason for all the spectra to not start at the origin. A possible source for the offset is that the electrolyte is reacting with the gold contact materials. The formation of such Na-Au alloys is possible at the temperatures reported here, so the authors should check if the electrodes have been affected by the measurement conditions.

The other explanation for the offset of the blue semicircle is that this corresponds to the grain boundary resistance of the sample. Since the samples tested are not dense (the authors report pellet densities in the range of 70-77%), it is likely that grain boundary resistances are the main contributor to the impedance spectra. If the explanation for the offset is the second one, then it is a big problem for the manuscript because the authors make direct comparisons of the theoretically obtained ionic conductivities and those obtained experimentally. Such a direct comparison is only possible with fully dense samples, impedance data single crystals or when the impedance data is analyzed considering grain (bulk) and grain boundary resistivities.

Since the impedance data analysis is flawed and it serves to support one of the key findings of the manuscript, namely “we have demonstrated that first principles- based kMC can access significantly longer time scales in the millisecond range and larger length scales (a total of 21,504 atom supercells in the present work), enabling us to approach experimental time and space resolutions and establish a robust link between measurements and theory”, I cannot recommend the manuscript for publication as is.

We appreciate the very valuable comment from the referee, which has enabled us to significantly improve the quality of this study. A similar comment was also raised by referee #2. As a consequence, we have performed several new additional measurements. The revised manuscript and the supporting information contain the description of the analysis of all these new experiments, which we briefly articulate below.

First of all, we have re-sintered our samples to reach much higher levels of compacity (> 90%) at $\text{Na}_{2.5}\text{Zr}_2\text{Si}_{1.5}\text{P}_{1.5}\text{O}_{12}$, $\text{Na}_3\text{Zr}_2\text{Si}_2\text{PO}_{12}$ and $\text{Na}_{3.4}\text{Zr}_2\text{Si}_{2.4}\text{P}_{0.6}\text{O}_{12}$, which helps to further reduce the effect from grain-boundary resistance on the total ionic conductivity. The characteristics of the new samples are shown in **Table 2**.

Table 2 Pellet compacity of $\text{Na}_{1+x}\text{Zr}_2\text{Si}_x\text{P}_{3-x}\text{O}_{12}$ re-sintered samples.

x	Compacity (%)
1.5	90.7
2.0	90.4
2.4	95.8

To decouple the grain boundary and bulk contributions to the total conductivity, we have conducted additional low temperature and high-frequency impedance spectroscopy

measurements on the re-sintered samples of **Table 2**, and as shown in **Figure 1** here (and **Figure 4b** of the main text, **Figures S12, S13** and **S14** of the SI). We have also re-measured the high-temperature impedance data of the high compacity pellets.

Low temperature (high frequency) data in **Figure 1**, e.g., 193 K, now clearly shows the closing semi-circle at $\sim 0 \Omega$ and the apparent grain-boundary contributions at higher frequencies. At higher temperatures (panel d-f **Figure 1**) these features are clearly lost.

Figure 1 Variable temperature impedance spectra of $\text{Na}_{1+x}\text{Zr}_2\text{Si}_x\text{P}_{3-x}\text{O}_{12}$ with nominal compositions (a, d) $x = 1.5$, (b, e) $x = 2.0$ (c, f) $x = 2.4$, respectively. The impedance spectra is obtained for different temperatures of ~ 193 K, ~ 273 K and 313 K and high-to-mid frequency for panel a-c and ~ 373 K, ~ 473 K and 573 K for mid-to-low frequency panel d-f. The equivalent circuits are shown above each plot. Examples of fits of the impedance spectra are shown in panels a, b and c. The bulk and apparent grain boundary resistances are qualitatively indicated by the semicircles and arrows as a guide for the eye.

From the impedance spectra we have extracted the activation energy from the total ionic conductivities vs. temperature, but also the bulk and grain boundary contributions in three representative NaSICON compositions, i.e., $\text{Na}_{2.5}\text{Zr}_2\text{Si}_{1.5}\text{P}_{1.5}\text{O}_{12}$, $\text{Na}_3\text{Zr}_2\text{Si}_2\text{PO}_{12}$ and $\text{Na}_{3.4}\text{Zr}_2\text{Si}_{2.4}\text{P}_{0.6}\text{O}_{12}$.

As temperature increases in **Figure 2**, the grain boundary conductivities approach monotonically (or show larger values in $\text{Na}_3\text{Zr}_2\text{Si}_2\text{PO}_{12}$) the values of bulk conductivities, and their differences appear negligible for any measurement above 333 K ($\sim 60^\circ\text{C}$). Therefore, it

is safe to say that at temperatures above 333 K (~60 °C) the total ionic conductivity is dominated by Na-ion transport in the bulk NaSICON.

Figure 2 Arrhenius plot of conductivity versus the reciprocal of temperature for $\text{Na}_{1+x}\text{Zr}_2\text{Si}_x\text{P}_{3-x}\text{O}_{12}$ with (a) $x = 1.5$, (b) $x = 2.0$ (c) $x = 2.4$ as extracted from high-frequency AC impedance spectroscopy. Bulk and apparent grain boundary (GB) contributions to the total ionic conductivities are shown separately.

As per **Figure 4b** of the main text, the highest regimes of ion transport in NaSICON is achieved at temperature above ~150 °C beyond its phase transition from monoclinic-to-rhombohedral for compositions $1.8 < x < 2.5$. We hope that the referee is well aware that separating bulk from grain boundary phenomena at high temperatures is not a trivial task, often approaching the limiting capabilities of high-frequency impedance spectrometers. Furthermore, our low temperature (< 60 °C) impedance measurements do indicate that the total conductivity is dominated by the bulk at elevated temperatures.

Because for NaSICON compositions with $1.8 < x < 2.5$ show a phase transition from monoclinic-to-rhombohedral, a change in slope of the Arrhenius curve is observed at high temperature (see **Figure 4b**). In contrast, at composition $x < 1.8$ and $x > 2.5$, $\text{Na}_{1+x}\text{Zr}_2\text{Si}_x\text{P}_{3-x}\text{O}_{12}$ remains in the rhombohedral phase. To this end, we could only extrapolate the grain boundary effects at temperatures higher than ~300 K using the impedance data for the $\text{Na}_{2.5}\text{Zr}_2\text{Si}_{1.5}\text{P}_{1.5}\text{O}_{12}$ composition, which is guaranteed to remain rhombohedral in the full temperature range (-100 to 300 °C). **Figure 3** demonstrates that the grain boundary contributions to the total conductivity is insignificant.

Figure 3 High-temperature extrapolation of bulk and grain-boundary contributions to the total ionic conductivity in $\text{Na}_{2.5}\text{Zr}_2\text{Si}_{1.5}\text{P}_{1.5}\text{O}_{12}$.

Note that we are not the first to demonstrate that the impact of grain boundaries on total Na-ion conductivity at high temperatures is trivial. See below for examples.

- Park et al. [ACS Appl. Mater. Interfaces, 2016, DOI: 10.1021/acsami.6b09992] had shown that the dominant factor determining the total Na-ion conductivity of NASICON switched from grain boundaries as a function of temperature ($< 100\text{ }^\circ\text{C}$) to bulk ($> 100\text{ }^\circ\text{C}$). This is in line with our data, which is on a larger set of Na compositions and temperatures.
- Lunghammer et al. [Chem. Phys. Lett., 2018, DOI: 10.1016/j.cplett.2018.04.037] showed that below room temperature, the effect of grain boundary in NaSICON on Na-ion conductivity is less than 0.5 orders of magnitude. Above room temperature, the grain boundary contribution has only little effect on the total conductivity.

Our new experiments together with the studies mentioned above convincingly suggest that grain boundary conductivity might influence the total Na conductivity only at sub-room temperatures and not at temperatures above $60\text{ }^\circ\text{C}$ ($100\text{ }^\circ\text{C}$, $200\text{ }^\circ\text{C}$ and $300\text{ }^\circ\text{C}$), which are the focus of this manuscript.

In accordance with these new measurements, the manuscript and the supporting information have been revised. A description of the measurement setup, impedance fitting and rationale behind the choice of the equivalent circuits have been discussed.

Answers to Reviewer #2

In this manuscript, the authors report the theoretical systematic study of the Na NASICON family by a novel kinetic Reverse Monte Carlo approach to assessing the impact of PO₄ substitution by SiO₄ polyanion. The authors also report experimental work on different compositions to validate their computation predictions. Overall, the work is well executed and of interest to the wider battery materials community and readership of Nature Communications, however, this reviewer suggests the below points to be addressed before publication of this work in Nature Communications:

We appreciate the referee's comments which have contributed to improve appreciably the quality of the manuscript. It is worth mentioning that our theoretical framework is called kinetic Monte Carlo and not "kinetic reverse Monte Carlo", which we believe is a different methodology.

Comment 1, Referee #2. The authors should discuss the limitation of their method for long-range diffusion analyses since the method does not incorporate the effect of grain boundaries. This is of particular interest when trying to understand experimental results where the material's microstructure can dominate the transport properties. Specifically in their work, where the authors compare their bulk (intra-grain) predicted transport properties with their experimental results, which seems to include both, bulk and grain boundaries (inter-grain) conductivity results. The authors could measure their transport properties at lower temperatures, which usually helps to decouple bulk and grain boundary resistances to Na⁺ transport.

The referee brought out an important point, which is similar in nature to comment #2 from Referee #1. We have amply addressed this comment above. In summary, i) the NaSICON pellets were re-sintered ensuring compacities >90%. ii) As suggested by the referee, using the new NaSICON pellets, **low-temperature and high-frequency AC** impedance measurements were performed to deconvolute the contributions of bulk and grain-boundary to Na-ion transport in NaSICONs (see **Figures 1, 2 and 3** of this document). With these measurements, we have carefully demonstrated that the effect of grain boundaries on Na-ion transport in NaSICON is negligible for temperatures larger than 60 °C (see **Figure 2**). Hence, we believe that our model captures the Na-ionic transport within the bulk NaSICON quite accurately, in agreement with our experimental bulk Na conductivity data.

The revised manuscript and SI includes several additional figures and text explaining our measurements and the resultant data.

Comment 2, Referee #2. This would be highly important, given that the authors have employed different synthetic methodologies and obtained different relative densities, which impacts the total conductivity reported. SEM images should be included to compare if microstructures are still comparable.

We thank the referee for the excellent comment. We have taken SEM measurements of the re-sintered pellets, i.e., Na_{2.5}Zr₂Si_{1.5}P_{1.5}O₁₂, Na₃Zr₂Si₂PO₁₂ and Na_{3.4}Zr₂Si_{2.4}P_{0.6}O₁₂. An example of the SEM analysis on Na_{3.4}Zr₂Si_{2.4}P_{0.6}O₁₂ is shown in **Figure 4** (see below). From the analysis of the SEM results, we could not observe significant variations in grain sizes (0.028 ~ 0.043 μm²) across the NaSICON compositions considered, which enables us to directly compare the effects of grain boundaries in the impedance analysis of the re-sintered

pellets. Note, we have implemented our code to do an determine an accurate statistics of grain sizes.

Figures S9-S11 have been added in the revised SI, as well as complementary sentences in the results and methodology sections of the main text.

Figure 4 Average grain area for $\text{Na}_{3.4}\text{Zr}_2\text{Si}_{2.4}\text{P}_{0.6}\text{O}_{12}$ was found to be $\sim 0.035 \mu\text{m}^2$ with the maximum grain area of $28.5 \mu\text{m}^2$ and minimum grain area of $4 \times 10^{-4} \mu\text{m}^2$.

Comment 3, Referee #2. The reported kMC method only allows for Na^+ to move, constraining the rest of the atoms to fix positions. Considering that for other polyanion systems have been reported polyanion rotations during Na^+ diffusion – so-called paddle-wheel effect, do the authors anticipate any limitation or their current methodology to this particular system or other sulphide systems where they expected to extend this methodology?

The framework of NASICON is three dimensional in nature, which is realized by corner-sharing of $(\text{Si}/\text{P})\text{O}_4$ tetrahedra and ZrO_6 octahedra. Compared to other electrolyte structures presenting “isolated” anion units, e.g., Na_3PS_4 , argyrodites, etc., the NASICON framework is to be considered as a “rigid” structure. This eliminates the possibility of paddle-wheel effects.

Also, as elaborated in Comment 1 of Referee #1, our model fully captures local structural distortions of octahedral and tetrahedral units in the NaSICON. Note that our first-principles calculations do not constrain the presence/absence of local distortion effects, and as a result

our calculated migration barriers fully include such effects. The local cluster expansion, which is fitted to the calculated barriers, is thus designed to coarse grain over the local distortions. Therefore, the kinetic Monte Carlo simulations, which rely on the local cluster expansion for calculating probabilities of hops, implicitly accounts for any local distortions within the structure. So, even if paddle-wheel effects were to play an important role, our model is designed to capture such effects.

This aspect has been clarified on Page 11 of the revised text of the main manuscript.

Comment 4, Referee #2. For the NBE method the authors have employed a temperature of 0 K, while with other methods and techniques temperatures above 300 K have been used. Do the authors expect a constant Arrhenius behaviour of the transport properties over the whole range of temperatures reported to make these results comparable?

The referee has raised an important point. First of all, we observe that Arrhenius behavior is typically obeyed by NaSICON structures (**Figure 4b**) and other oxide ion conductors, e.g., $\text{La}_7\text{La}_3\text{Zr}_2\text{O}_{12}$ [see Allen et al., 2012, DOI: 10.1016/j.jpowsour.2012.01.131]. 0 K NEB calculates the energy barrier for a single independent Na hop within a single phase, which forms the Arrhenius slope.

Activation barriers do depend on vibrational properties of the rigid framework, which depends (weakly) on temperature. Previously, Vineyard and Van der Ven [DOI: 10.1016/00223697(57)90059-8; 10.1103/PhysRevB.64.184307] have proposed strategies to account for vibrational entropy in the vibrational prefactor of the Arrhenius expression for diffusivity. However, modelling such effects is challenging even for single species materials or alloys. Thus, it is usually a reasonable approximation that activation barrier calculated at 0 K holds for higher temperatures also for a given hop.

Experimentally, energy barriers are extracted by fitting the measured conductivity vs. temperature using the Arrhenius relationship, probing an ensemble of possible Na migrations during measurement. Different hops will be activated at different temperatures, and the “effective” hop(s) (i.e., statistical average of active hops) will represent an “effective” barrier (statistical average of barriers).

Within the kinetic Monte Carlo (kMC) framework, we associate a probability of occurrence for

each hop, which is proportional to $e^{-E^{\ddagger}/k_B T}$, where E^{\ddagger} is the barrier of the hop, k_B is the Boltzmann constant and T is the temperature. Subsequently, we determine the effective hop(s) by “selecting” hops stochastically and “averaging” over such hops. Since E^{\ddagger} is a weak function of temperature, whether a hop contributes to the effective hop (and the effective barrier) is largely determined by the temperature at which the kMC simulation is performed. Hence, **kMC provides a robust and accurate estimate of ionic conductivities as a function of temperature via stochastic sampling of an ensemble of possible hops.** Therefore, we believe that our 0 K NEB calculations, used along with our kMC model should be able to mimic the statistical nature of Na-transport in NaSICON and other ionic conductors as well as electrode materials.

Comment 5, Referee #2. The authors use the structure obtained at 1500 K to mimic that of the maximum temperature employed during experimental synthesis. In most cases, however, materials are cooled down to ambient temperature naturally, allowing the crystal structure to relax. This also seems to be the case of their experimental condition – no quenching protocol is mentioned. The authors should discuss this point and recalculate it as necessary.

Thank you for this important comment. A canonical Monte Carlo at 1500 K, which explores the thermodynamic landscape of the NaSCION structure, before each kMC simulation is simply required to ensure that silicate and phosphate units are randomly distributed in all the model supercells, as observed in structural analysis (XRD/Neutron diffraction experiments) of this material. Previously, we have also demonstrated that Si/P redistribution upon cooling is hindered due to the high migration energy barriers (~4.02 eV) of SiO₄ and PO₄ units [Deng et al. 2020, DOI: 10.1021/acs.chemmater.0c02695].

The revised manuscript includes two clarifying sentences on Page 12 of the main text.

Comment 6, Referee #2. Additionally, the authors must perform Rietveld refinements, rather than Le Bail fittings, to obtain full structural information, importantly Na distribution within Na(1) and Na(2) sites, stoichiometry as well as quantification of ZrO₂ impurity presence. Particularly, the presence of this impurity is concerning in their Na = 3 sample, where a non-negligible amount of ZrO₂ is detected, which could affect the total transport properties of the material. Additionally, the authors' claim on phase purity of these compounds should be revised.

Thank you for this important comment. As requested by the referee, we have performed Rietveld refinements that are presented in the revised **Figure 4a**, **Tables S7, S8** and **S9** in the SI contain the Na occupancies, the atomic coordinates, refined lattice constants of the NaSICON structures obtained from Rietveld refinements. We have also quantified the ZrO₂ content of all NaSICON compositions, as reported in **Table S6** of the revised SI. The highest ZrO₂ content was found in Na_{3.0}Zr₂Si₂PO₁₂ (~10 wt%) and the lowest (~0.1 wt%) in Na_{3.4}Zr₂Si_{2.4}Po_{0.6}O₁₂. We agree with the referee that the ZrO₂ content in Na_{3.0}Zr₂Si₂PO₁₂ is not insignificant as in the case in Na_{3.4}Zr₂Si_{2.4}Po_{0.6}O₁₂. Nevertheless, our measured ionic conductivities in Na_{3.0}Zr₂Si₂PO₁₂ are comparable or better than data available in the literature. Furthermore, the referee should also note that the novelty of the manuscript is the original computational method and not the synthesis of known NaSICON compositions. We synthesized these NaSICONs compositions in order to measure their ionic transport for comparison with our predictions. Previously, it has been speculated that ZrO₂ formation lowers the Zr content in the main NASICON phase and presumably leading to an enrichment of Si and Na at the grain boundaries and evidently influencing the conductivity. However, our Rietveld analysis does not indicate any loss of Zr. This is accompanied by the impedance data measured in this manuscript, which indicates favourable grain boundary effects on the total Na-ion conductivity.

Comment 7, Referee #2. It would be helpful for the reader to include in the supporting information the structural parameters for the models employed.

While resubmitting this manuscript, we have included all the initial structures (cif format) of our kMC simulations in our data repository (https://github.com/caneparesearch/NASICON_KMC_paper_data/tree/main/CIF_structures)

containing the structural parameters requested by the referee. We have revised the sentence under the section "Code and Data Availability".

Comment 8, Referee #2. Considering these are quenched structures at 1500 K, which is the symmetry of their model used for the diffusion calculations? This is an important parameter considering the monoclinic to rhombohedral transition of the NASICON system at some compositions at temperatures below the quenching point selected. Additionally, Na⁺ occupancies across different sites could vary with temperature.

We used 1500 K to introduce disordering in Si/P. For this process, we used a rhombohedral structure, which is readily comparable with experiment. To take into account the effect of Na/Vacancy and Si/P disorder, as well as the structural variance in a statistical fashion, for each Na composition, 50 distinct NaSICON structure models were assessed.

Furthermore, the phase transition (rhombohedral-to-monoclinic) is mainly the result of disorder (rhombohedral) to order (monoclinic) transformations on the Na/Vacancy and Si/P sublattices. Our structural and computational model fully captures these order/disorder situations depending on conditions (temperature, composition). We have extensively documented these aspects in a previous publication [Deng et al. Chem. Mater., 2020, DOI: 10.1021/acs.chemmater.0c02695].

We hope this explanation clarifies the referee's doubts. The revised manuscript includes two clarifying sentences in the methodology (Page 32).

Comment 9, Referee #2. This reviewer also recommends the authors to calculate/show the windows sizes difference for Na⁺ diffusion when replacing P⁵⁺ with Si⁴⁺, and correlate these with their results/conclusions.

We acknowledge that different NaSICON structures with different transition metal and anion species, can affect the window sizes (bottlenecks) for Na⁺ migration (indicated as T1 and T2 in **Figure 5b**), with an appreciable effect on the migration barriers. These effects have been extensively documented [DOI: 10.1016/j.jpowsour.2014.09.137]. However, **Figure 5a** demonstrates that variation of window sizes has no direct correlation with the migration energies in the same NaSICON structure, i.e., Na_{1+x}Zr₂Si_xP_{3-x}O₁₂. Therefore, from **Figure 5a**, we cannot infer a direct correlation between the size of the migration bottlenecks (indicated as the area T1 and T2) and the diffusion barrier when the P⁵⁺ and Si⁴⁺ content is varied.

Indeed, this study and our previous work [DOI: 10.1021/acs.chemmater.0c02695] demonstrated that variation Si/P has more impactful effects on order-disordering phenomena, which are deemed more important than bottleneck sizes.

Figure 5 Panel a shows the bottleneck areas as function of Si/P content in the NaSICON structure and computed barriers. **Panel b** defines T1 and T2, which indicates the areas of the triangular bottlenecks for Na-ion migration. Barriers at $x = 2$ are taken from different Si/P local environments.

Comment 10, Referee #2. The authors should include in the abstract the temperature at which the reported conductivity is obtained. Furthermore, the authors claim high conductivity (page 28) of their sample with 0.1 mS cm^{-1} at 200 oC, considering the current state-of-the-art in the field this claim should be revised.

Thanks for this comment. We have corrected this imprecision in the abstract of the revised manuscript. The new experimental measurements on denser pellets (0.165 S/cm at 473 K) are in much better agreement with simulations (0.170 S/cm at 473 K). The referee should note that the units are S/cm not mS/cm.

Comment 11, Referee #2. The authors should also consider including the effect of the higher or optimal Na stoichiometry when discussing the positive effect of P⁵⁺ substitution with Si⁴⁺ on page 26.

We thank for the comment. On page 30 of the revised manuscript a new sentence has been added “For NASICONs with higher Si content, Na off-stoichiometry achieved by “stuffing” excess sodium in the material might provide increased Na mobility. This strategy is commonly applied in many ion conductors [Ref]”

Comment 12, Referee #2. As an additional comment of this reviewer, and out of the scope of the current work, this reviewer finds quite interesting and encourage the authors to investigate coupling their methodology to model the disorder or phase segregation with total scattering techniques to shed light on any clustering of this mixed polyanion systems that could escape conventional structural characterisation.

We thank the referee for the excellent comment. We have included a sentence on page 28 of the revised manuscript: “The analysis of Error! Reference source not found. can also be relevant for the rationalization of complex total scattering experiments obtained, for example by neutron-based techniques.”

Dr Marco Amores, Austrian Institute of Technology

Answers to Reviewer #3

In this paper, Deng et al. performed both experimental and theoretical simulations to elucidate Na ion diffusion properties both in bare $\text{NaZr}(\text{PO}_4)_3$ and Si doped on the P site in $\text{NaZr}(\text{PO}_4)_3$. This is a systematic study. However, I have a question in its novelty. There are many studies (hypothetical and experimental) available showing that additional Li or Na can be produced in many cathode materials [ex: Recently, Arunkumar et al. synthesized over-lithiated $\text{Li}_{2+x}\text{Ru}_{1-x}\text{CoO}_3$ cathode by aliovalent Co doping on Ru site in Li_2RuO_3 and concluded that there is an enhancement in the electrochemical lithium reversibility and Li^+ extraction compared to those associated in the pristine Li_2RuO_3 [Arunkumar, P., Jeong, W. J., Won, S. & Im, W. B. Improved electrochemical reversibility of over-lithiated layered Li_2RuO_3 cathodes: Understanding aliovalent Co^{3+} substitution with excess lithium. J. Power Sources 324, 428–438 (2016)]. This enhances the capacity and diffusion. Here, authors choose a phosphate based material to show that there is an agreement with theory and experiment. Editor should decide whether this manuscript is considered or not. If yes, the following issues should be considered

We thank the referee for his/her time in revising this manuscript. We respectfully disagree with this comment. We clarify the reasons of our disagreement below:

1. Here, we are dealing with a solid electrolyte, not a cathode materials. In fact, Zr^{4+} is a closed shell (d^0), non-redox-active transition metal (under typical electrochemical conditions). The material studied here, NaSICON is a complex polyanion material and not a “simple” ternary oxide as mentioned by Reviewer #3.
2. In this comprehensive study on the NaSICON electrolyte, we deal with the Si/P disorder over the entire Na^+ composition range, instead of a mere doping task as suggested by Reviewer #3. This manuscript elucidates the complex mechanisms behind Na-ion transport in NaSICON materials as a function of temperature and Na content.
3. As a major outcome of this manuscript, we propose a novel strategy to improve ion transport in NaSICON-based solid electrolytes by replacing P with Si. **Unfortunately, the referee believes that our new findings are as trivial as adding an excess of cations** (increase Na concentration) as in $\text{Li}_{2+x}\text{Ru}_{1-x}\text{CoO}_3$. **Clearly, this is not the case in our study.**
4. Finally, this study represents the first non-trivial endeavor combining seemingly advanced theoretical techniques, such as, density functional theory, nudged elastic band, cluster expansion, and kinetic Monte Carlo to describe the complexity of Na-ion transport in solid electrolyte materials. This simulation framework enables the generation of long-time statistics of ionic transport, which are required to capture the correlation effects in Na-ion transport. **We are presently unaware of any computational modelling work on solid electrolytes developing/using this level of sophisticated theoretical tools.** Our simulation method can approach unprecedentedly long (ms) time sampling on extremely large structure models (with more than 10,000 atoms).

In summary, we believe the referee has grossly misjudged the novelty of our work. In contrast, Referees #1 and #2 could grasp this novelty.

Comment 1 Referee #3: Nonetheless, for $\text{Na}_3\text{Zr}_2\text{Si}_2\text{P}_1\text{O}_{12}$ our measured experimental conductivity (see Figure 4b) of $\sim 0.1 \text{ S cm}^{-1}$ at 573 K is in excellent agreement with our computed value of $\sim 0.2 \text{ S cm}^{-1}$. Clearly calculated value is overestimated by 100%. Is that excellent agreement?

The order of magnitude error of the reported measurements is in the order of ~ 10 . This is supported by large dataset of measurements (spanning over several orders of magnitude) recorded at similar temperature and Na compositions in the literature, as shown **Figure S6b** of the SI. Clearly, the order of the magnitude of the error is important, and Referee #3 should not just simply compare the computed number with the measured values. Furthermore, our new experimental measurements on denser pellets at $x = 2.4$ (0.165 S/cm at 473 K) are in much better agreement with our kMC simulations (0.170 S/cm at 473 K).

The updated **Figure 2b** demonstrate the large variability of measured ionic conductivity for the same samples but different level of densification (achieved through different sintering regimes).

Comment 2 Referee #3 Structures with charge-neutral vacancies at initial and final images in the diffusion path were fully optimized until the interatomic forces were less than 0.01 eV/\AA and stresses were less than 0.29 GPa . Stress tensor looks high, not zero. Normally it should be 0.05 GPa or less than this.

We thank the referee for the comment. Indeed, there is no flag to control the convergence of the stress tensor in VASP. We have removed this confusing sentence from the SI on Page S4. Indeed, the computed stress tensors are below 0.05 GPa .

REVIEWERS' COMMENTS

Reviewer #1 (Remarks to the Author):

I commend the authors for their effort in the revisions of the manuscript. They have addressed my initial concerns/comments in such a way that I can recommend this manuscript to be published as is.

Reviewer #2 (Remarks to the Author):

The authors have successfully addressed the comments raised by this reviewer, including sufficient additional experimental work and analyses, thus I recommend publication of this revised manuscript in Nature Communications.

Dr Marco Amores, Eurecat - Technology Centre of Catalonia

Reviewer #1 (Remarks to the Author):

I commend the authors for their effort in the revisions of the manuscript. They have addressed my initial concerns/comments in such a way that I can recommend this manuscript to be published as is.

We are grateful to read that Reviewer #1 fully support our manuscript for publication in Nature Communication.

Reviewer #2 (Remarks to the Author):

The authors have successfully addressed the comments raised by this reviewer, including sufficient additional experimental work and analyses, thus I recommend publication of this revised manuscript in Nature Communications.

Dr Marco Amores, Eurecat - Technology Centre of Catalonia

We are grateful to read that Reviewer #2 fully support our manuscript for publication in Nature Communication.